# Insights into natural neocentromere evolution from a cattle T2T X chromosome

Paulene S. Pineda[1,6], Callum MacPhillamy[1,6], Yan Ren[1,6], Tong Chen [1], Luan Zhong[2], David L. Adelson [2], Carey Dessaix[2], Jose Perez-Silva [3], Leanne Haggerty[3], Fergal J. Martin[3], Cynthia D. K. Bottema[1], Wayne S. Pitchford[1], Benjamin D. Rosen [4], Timothy P. L. Smith [5] & Wai Y. Low [1] ✉

The cattle genome is crucial for understanding ruminant biology, but it remains incomplete. Here we present a telomere-to-telomere haplotype-resolved X chromosome and four autosomes of cattle in a near-complete assembly that is 431 Mb (16%) longer than the current reference genome. Using this assembly (UOA_Wagyu_1) we identify 738 new protein-coding genes and support the characterization of centromeric repeats, identification of transposable elements, and enabled the detection of 2397 more structural variants from 20 Wagyu animals than using ARS-UCD2.0. We find that the cattle X centromere is a natural neocentromere with highly identical inverted repeats, no bovine satellite repeats, low CENP-A signal, low methylation, and low CpG content, in contrast to the autosomal centromeres that are comprised of typical bovine satellite repeats and epigenetic features. Our results suggest it likely formed from transposable element expansion and CpG deamination, suggesting dynamic evolution. We find eighteen X-pseudoautosomal region genes have conserved testes expression between cattle and apes. We also find all cattle X neocentromere protein-coding genes are expressed in testes, which suggests they potentially play a role in reproduction.

The initial assembly of the bovine genome was reported in 2009[1]. Multiple assemblies have since been generated, each one providing new insight into the genomic structure of cattle[2-7]. However, long stretches of biologically important repetitive sequences such as the ribosomal DNA (rDNA), telomeres and centromeres[8-10] as well as satellite sequences continue to challenge complete telomere-to-telomere (T2T) assembly. Allosomes generally contain a higher proportion and density of repetitive sequences, making them more difficult to assemble than autosomes until recently[3,11,12]. The X chromosome is important as it contains several genes with important roles in a multitude of traits, such as intelligence and mental function[13], intellectual disability[14,15], reproduction[16,17], and milk production[17]. The

lack of complete sex chromosome assemblies has hampered efforts to understand their role in these traits. Over 20 *Bos taurus* assemblies are presented in the NCBI database currently, but none of these previous assemblies contain a complete X chromosome.

The assembly of five complete T2T chromosomes for *Bos taurus* in UOA_Wagyu_1 provides an opportunity to fully characterize bovine centromeres and contrast autosomal and allosomal centromere features. Centromeres are essential for the correct partitioning of chromosomes during mitosis and meiosis[18]. They are typically characterised by tandem repeats organized into higher-order repeats (HOR)[19,20], enrichment of the centromere protein A (CENP-A) marking the site of kinetochore assembly[21], and dense methylation[22], except for

[1]Davies Livestock Research Centre, School of Animal and Veterinary Sciences, University of Adelaide, Roseworthy, SA, Australia. [2]School of Biological Sciences, University of Adelaide, Adelaide, SA, Australia. [3]European Molecular Biology Laboratory, European Bioinformatics Institute, Wellcome Genome Campus, Hinxton, Cambridge, UK. [4]Animal Genomics and Improvement Laboratory, ARS USDA, Beltsville, MD, USA. [5]US Meat Animal Research Center, ARS USDA, Clay Center, NE, USA. [6]These authors contributed equally: Paulene S. Pineda, Callum MacPhillamy, Yan Ren. ✉e-mail: wai.low@adelaide.edu.au

regions of hypomethylation within CENP-A core domains[23]. In rare cases, the centromere can reposition to an ectopic location, resulting in the formation of a neocentromere[24–26]. Two types of neocentromere formation have been described: the first type has been observed in human clinical cases and is known as the human neocentromere (HN)[27]. HNs can lead to severe diseases such as congenital abnormalities and cancer[27]. Moreover, unlike canonical centromeres, HNs do not contain the alpha satellite DNA and are only found in single copy sequences[27,28]. However, like canonical centromeres, they are still identified by their CENP-A signal; although low levels are found along the chromosome[29]. Given their often deleterious nature, HNs are not fixed within populations. The second type has been observed in eight mammalian species (horse[30], donkey[31], plains and imperial zebra[32], Bornean and Sumatran orangutan[33], squirrel monkey[34], and macaque[35]) and are believed to have arisen via a non-deleterious centromere relocation that has then become fixed within the population[34]. This type of neocentromere is known as an "evolutionarily new centromere" (ENC). These ENCs are generally characterised by either tandem repeats or non-repetitive sequences and are associated with CENP-A[25,34]. The methylation status of these ENCs is not known.

As new reference genomes become available, it is important to compare their performance in population genetics analyses like variant calling compared to current reference genomes. Recent work with human T2T genomes has revealed substantial reference bias when mapping East Asian samples to CHM13 compared to the East Asian-specific CN1 genome, with 6641 structural variants (SVs) present in CN1 but absent in CHM13[36]. These SVs represent a substantial degree of missing variation due to reference genome choice. The current cattle reference genome is based on the Hereford breed[2] that does not capture all cattle genetic diversity and still consists of many gaps, which prevents unbiased SVs discovery. This work creates a new reference genome for Wagyu and is an opportunity to compare SV calling results against the cattle reference ARS-UCD2.0.

We present a gapless T2T X chromosome for cattle (BTAX) in an assembly also containing four complete autosomes. We present detailed analyses of these new sequences, with particular emphasis on BTAX and its centromere and present evidence of it being a natural neocentromere. We then demonstrate the utility of this new assembly by generating a catalogue of cattle structural variants (SVs) from 20 unrelated Wagyu samples. With the assembly of complete sex chromosomes for human[8,37], apes[38], sheep[39] and goat[40], and now cattle[41], we have an immensely valuable resource for studying sex chromosome evolution.

## Results

### Genome assembly

A male calf from a Wagyu dam and Tuli sire cross was selected for haplotype-resolved genome assembly. We generated 228.8x ONT long-read coverage including 18.3x ONT ultra-long (>100 kb), 58.1x PacBio HiFi, ~421 million Proximo Hi-C reads, 81.8x F1 Illumina short-read, and 78x parental Illumina short-read coverage (Supplementary Data 27). Ten assembly versions were created using two assembly algorithms, Verkko[42] and hifiasm[43], and combinations of HiFi reads, ONT reads and HERRO corrected ONT reads input sequence data to achieve the most contiguous genome assembly that have the most T2T chromosomes (Supplementary Data 1, Supplementary Note 1-2). The sequencing coverage of the assembly selected was 58.1x of PacBio HiFi reads, 57x of HERRO corrected ONT reads as input along with HiFi, 121x coverage of ONT reads and 18.3x ONT ultra-long reads (>100 kb). The Wagyu haplotype from the initial Verkko assembly was selected for further curation based on quality value (QV) metric, overall contiguity, and the number of chromosomes with telomeric sequence at both ends (Supplementary Note 1). The initial assembly contained 425 contigs with a contig N50 of 111.1 Mb and only 3.5% of the total bases were in unplaced scaffolds after filtering the contigs. The final assembly after curation consisted of 419 contigs with similar contig N50 as before since the removed contigs were small. Polishing the draft assembly with PacBio HiFi reads and DeepVariant slightly increased the overall QV from 54.30 to 54.43 (Supplementary Data 2). The final genome assembly for Wagyu was 3.14 Gb in length with 17 gaps and had a 99% BUSCO completeness score (Table 1). There were five T2T chromosomes, including the four autosomes BTA9, BTA10, BTA21, and BTA23, and a complete bovine X chromosome. The assembly had 431 Mb (16%) more bases than the bovine reference assembly ARS-UCD2.0 (GenBank accession GCF_002263795.3[2]), excluding the reference Y chromosome (Fig. 1a). A total of 20 chromosome-length contigs were gapless, but 16 of them did not have telomeres on both ends and some did not span the centromeres (Fig. 1b). Approximately 86.8% of the unplaced scaffolds consisted of repeats (Fig. 1c). The assembly graph of UOA_Wagyu_1 illustrated that the principal cause of the majority of autosomes being incomplete was due to unresolved tangles in the graph. These tangles typically occurred at the ends of the p-arms and involved satellite repeats in the centromeric regions of acrocentric autosomes (Fig. 1d, e). Five chromosomes also had tangles related to tandem rDNA units at the distal ends of their q-arms (Fig. 1f). The genome annotation was performed by EBI using Ensembl Gene Annotation system, which integrated both short- and long-read transcriptomic data with protein homology model, with emphasis for gene models with transcriptomic data support (Supplementary Note 3).

### Repeat annotation and rDNA copies

The UOA_Wagyu_1 assembly is made up of 51% repetitive sequences, a substantial increase compared to the 41% in ARS-UCD2.0 (Fig. 1g; Supplementary Data 3). Incorporating Pacific Biosciences (PacBio) HiFi and Oxford Nanopore Technologies (ONT) ultra-long (UL) reads in the UOA_Wagyu_1 assembly improved the resolution of the repetitive regions, expanding the centromeric satellite repeats to 12% of the genome from the 2% in ARS-UCD2.0 and accounting for the bulk of the increased genome size.

The human 45S rDNAs are 43 kb long, near-identical tandem repeats that posed a challenge to generate T2T genome assemblies[8]. In Wagyu cattle, nine complete rDNA copies were identified on four chromosomes, and 190 copies in 100 unplaced scaffolds, posing assembly challenges similar to the human genome[8]. Notably, rDNA

## Table 1 | Assembly statistics

| | Whole genome | Autosomes and X chromosome only |
|---|---|---|
| Genome Size | 3142409774 | 3031482482 |
| Contig N50 | 111091738 | 111091738 |
| Scaffold N50 | 114962053 | 116587974 |
| Number of contigs (+MT) | 419 | 30 |
| Number of scaffolds | 402 | 30 |
| Number of unplaced scaffolds | 371 | - |
| Mitochondrial genome | 1 | - |
| Size of unplaced scaffolds | 110910952 | - |
| Number of gaps | 17 | 12 |
| Merqury QV | 54.4331 | 60.6184 |
| Switch-error rate | 0.36 | 0.36 |
| Completeness (k-mers) | 99.9327 | 99.8388 |
| Compleasm (BUSCO) | S:98.78%, 13171 D:1.08%, 144 F:0.08%, 11 I:0.00%, 0 M:0.06%, 8 N:13334 | S:98.75%, 13167 D:1.10%, 147 F:0.09%, 12 I:0.00%, 0 M:0.06%, 8 N:13334 |

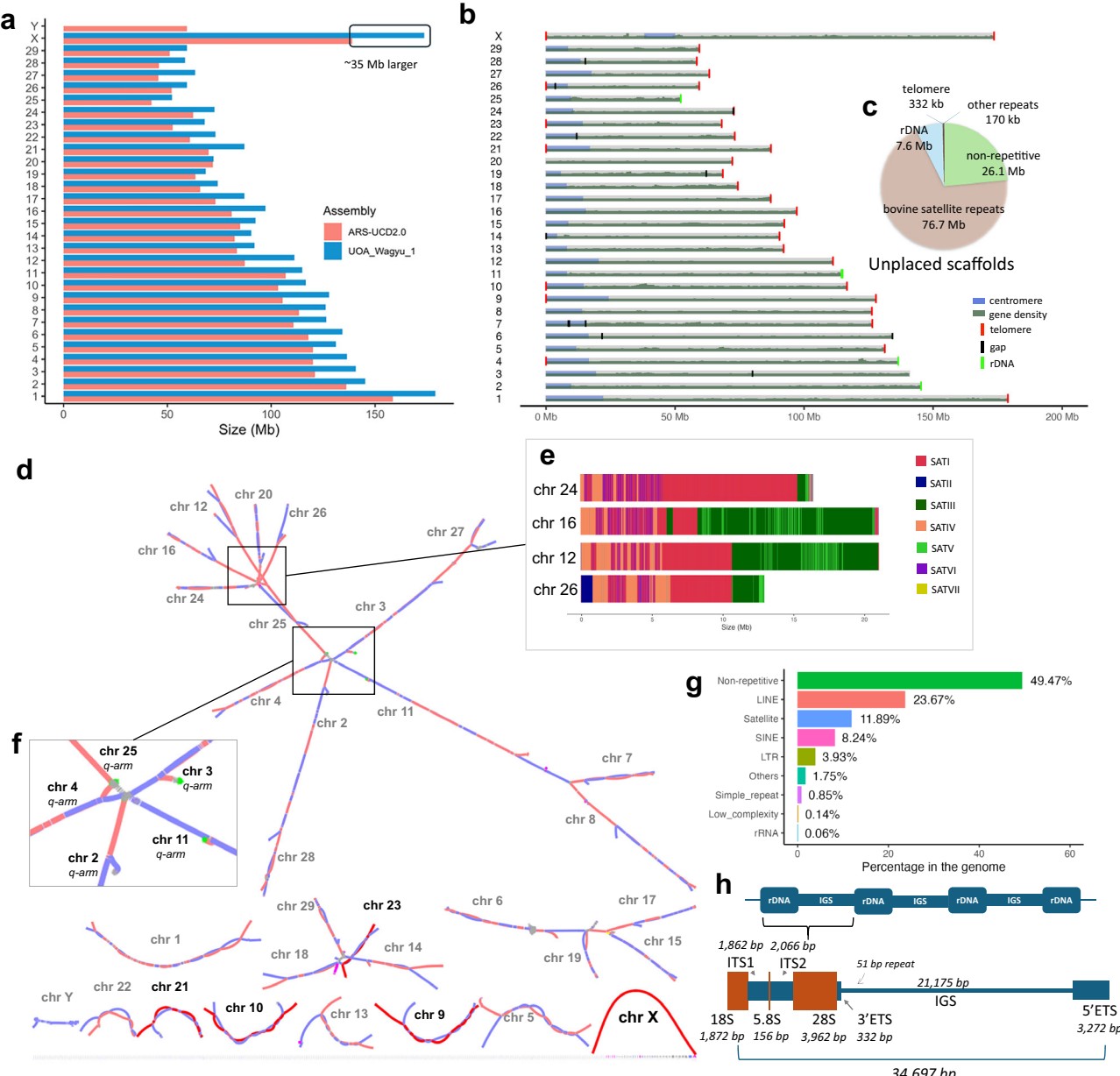

**Fig. 1 | Summary of the UOA_Wagyu_1 genome assembly. a** Chromosome size comparison between UOA_Wagyu_1 and ARS-UCD2.0 with the additional region highlighted in a boxed outline. **b** Ideogram of the UOA_Wagyu_1 genome assembly, indicating the location of the centromere (blue), assembly gap (black) and the telomeres (red), gene density (dark green) and ribosomal DNA (rDNA) (neon green). **c** Annotation of sequences in the unplaced scaffolds. **d** Assembly string graph visualized using Bandage of UOA_Wagyu_1 and UOA_Tuli_1 genome assemblies from Verkko. The nodes (strings) are color-coded as follows: pink for the maternal haplotype (UOA_Wagyu_1), blue for the paternal haplotype (UOA_Tuli_1), red for the T2T (telomere-to-telomere) Wagyu chromosomes, and green for nodes containing complete rDNA copies in Wagyu. **e** The centromeres of the four chromosomes sharing the same general structure with interspersed bovine satellite repeats SATI and SATVI, contributing to the tangle observed in the string graph. **f** A zoomed-in view of the rDNA tangle on the q-arm of five chromosomes, highlighting the unplaced scaffolds containing rDNA sequences (in neon green). **g** Repeat and non-repetitive content of the genome. **h** The genetic structure of the consensus rDNA unit that spans ~35 kb. There is a 51 bp repeat that starts at the 3'ETS (external transcribed spacer) region that differs in copy number in each rDNA unit. Source data are provided as a Source Data file.

arrays prevented the complete resolution of BTA4 (Supplementary Note 4). The cattle ~35 kb rDNA unit consists of 18S, 5.8S and 28S with ~99% sequence identity, and an intergenic spacer (IGS) region that varied substantially (Fig. 1h; Supplementary Data 4; Supplementary Note 4).

### Telomeres at the p and q-arms of chromosomes
As no cattle assembly has complete telomeres at both ends of chromosomes and analysis of telomeric units is lacking in most cattle assemblies, we assessed the completeness of assembled Wagyu chromosomes. We assembled five T2T chromosomes, one TgapT chromosome, 19 chromosomes with telomeric repeats at only one end and five chromosomes without telomeres at either end (Supplementary Data 5). The Wagyu genome has 64,965 telomere units whereas the current cattle reference (ARS-UCD2.0) only has 6849 telomere units (Supplementary Fig. 1a). There were 31 unplaced scaffolds with telomere repeats at their termini with a total size of 26 Mb. For assembled chromosomes in the genome, the average telomere length was 6.8 kb. The average telomere length from the PacBio HiFi reads was 3.1 kb (Supplementary Note 5).

## Genomic architecture of centromeric satellite repeats

Resolving the genomic architecture of cattle centromeres such as repeat type, the number of repeating units, sequence identity to each other and their organisation is crucial for assembling full chromosomes. The long tandem repeats in cattle acrocentric chromosomes have made it difficult to bridge the centromere to telomere in the p-arm. Specifically, the tangles in BTA12, BTA16, BTA24 and BTA26 were most likely attributed to the similarity of their centromeric structure (Fig. 1e). Nevertheless, five autosomes (BTA4, BTA9, BTA10, BTA21 and BTA23) were assembled, spanning the centromere (Fig. 2a; Supplementary Data 5). About 84% of centromeric regions in the genome were comprised of the seven bovine satellite repeats (SATs) (Fig. 2b). CENP-A enrichment analysis of the bovine satellite repeats showed highest level in SATIII in both CENP-A levels (Fig. 2c; Supplementary Data 6; Supplementary Note 6), indicating possible centromere function for this satellite repeat.

The locations and copies of the different bovine satellite repeats varied between chromosomes (Fig. 2d; Supplementary Data 7; Supplementary Note 7). However, all resolved autosomal centromeres have the following order from the p-arm to end of centromere: a telomere, followed by the SATVII with short interspersed nuclear element (SINE), long interspersed nuclear element (LINE) and simple repeats, SATII and SATIV (Fig. 2e). The most abundant bovine satellite repeats in the centromere were SATI and SATIII (Fig. 2b). The centromeric regions in autosomes could be classified into two structures based on the patterns of the satellite repeats, which was determined by the presence of interspersed SATI and SATVI (Fig. 2e, f; Supplementary Fig. 2).

## Inverted and sex-specific repeats in the BTAX centromere

There were several lines of evidence supporting the BTAX centromere occurring at position 38–50 Mb such as the presence of the previously reported bovine X centromeric marker (AJ884576)[44] and its submetacentric position (Supplementary Note 8). The BTAX centromeric region spanning 12 Mb was composed of 88.7% highly identical inverted repeats consisting of 72.2% transposable elements (TEs) (Fig. 3a–c; Supplementary Fig. 1b). The length of inverted repeats ranged from 10 bp to 620 kb (Fig. 3d; Supplementary Note 9). No bovine satellite repeats were found in the BTAX centromere that corresponded to the hallmarks of autosomal centromeres. Additionally, low CENP-A level was observed in comparison to the autosome centromeres.

The BTAX centromere displayed a repeat block structure with <90% sequence identity in moddotplot (Fig. 3c; Supplementary Fig. 1b). Within these blocks, we identified two repeats—2898 bp (111 copies) and 6558 bp (106 copies)—covering 8.5% of the centromeric region. Additionally, nine 318 kb repeats (99% identity) accounted for 24% of the centromeric region. We designate these as XCTR1 (2898 bp), XCTR2 (6558 bp), and XCTR3 (318 kb) (Supplementary Fig. 3). The XCTR1 contains AJ884576.1, a known BTAX centromere marker[44]. Additionally, two X specific repeats were found using comprehensive ab initio repeat pipeline (CARP)[45] which were a 550 bp repeat (family 0 consensus) with 23 copies and a 636 bp repeat (family 3 consensus) with 35 copies (Supplementary Fig. 4). We named these repeats as XCTR4 and XCTR5, respectively. The largest BTAX centromere TE from the RepBase (v29.04) library is Bisbis-1.8, covering a total of 849 kb of the BTAX centromere, followed by LTR11B_BT (477 kb) and BosInd-1.103 (300 kb)[46].

## The BTAX centromere exhibits a unique CpG profile

Centromeres are known to be epigenetically defined, with recent work revealing a centromeric dip region (CDR) where CpG methylation decreases and coincides with the presence of CENP-A[47]. To determine whether the BTA centromeres followed this pattern, we investigated DNA methylation and CENP-A profiles within UOA_Wagyu_1_Y. While high CENP-A and CDRs were not observed in BTAX centromere[47], there are interesting methylation patterns and CpG profiles between the

chromosomes. Based on UOA_Wagyu_1_Y (see Methods), centromeric regions had significantly lower median methylation than noncentromeric regions (Supplementary Note 10).

The greatest number of significant centromere vs noncentromere comparisons were at the 100 kb window size (Fig. 4a; Supplementary Data 8). In 54 of the 58 chromosome-wide comparisons, the centromeres exhibited significantly less methylation. BTAX had the lowest median centromeric methylation of all the chromosomes and the greatest methylation difference between the centromere and non-centromere (-10%).

The relationship between the observed CpG count and the expected CpG count was linear (Pearson's R = 0.95, P-value = 1.01e-18) (Fig. 4b) (Supplementary Methods). The BTAX centromere was the only outlier among the centromeres, with an observed CpG count far below what was expected given the number of C and G nucleotides in the centromere (Supplementary Data 9). Moreover, the proportion of the BTAX centromere that was comprised of CpG dinucleotides was -0.8% compared to -4–6% for the autosomal centromeres. Indeed, the ratio of CpG to non-CpG dinucleotides was significantly smaller in the BTAX centromere (0.02) compared to the autosome centromeres (0.07) (Fisher's exact test, P-value < 0.0001). Interestingly, TpGs occurred significantly more frequently within the BTAX centromere than CpGs (Fisher's exact test, P-value < 0.0001) and the ratio of observed to expected CpGs and observed to expected TpGs was the smallest of all centromeres (Supplementary Fig. 5). Together, these results suggest a depletion of CpGs relative to TpGs within the BTAX centromere. The correlation between the observed and expected CpG counts in UOA_Wagyu_1_Y differs substantially from the pattern observed in human (CHM13v2.0/hs1), where the correlation between observed and expected CpG counts was weaker in human than cattle (Fig. 4c). BTAX had the smallest proportion of CpGs within the centromere when compared against the X centromeres of primates (Supplementary Data 10).

## Dynamic evolution of X centromere

The X centromere displayed dynamic evolution between species from Hominidae and Bovidae families, with four different centromere positions found across humans, chimpanzees, cattle, water buffalo, sheep and goats (Fig. 4d). Sequences surrounding the X centromere of sheep (OARX) and goats (CHIX) were syntenic, which suggested their ancestral centromere position was similar. However, this ancestral X centromere position was not shared with cattle and water buffalo. While the water buffalo X chromosome (BBUX) is acrocentric like sheep and goats, its centromere position is unique and its X pseudoautosomal region (X-PAR) is on the q-arm, similar to cattle. Although the assembly of the water buffalo X centromere (BBUX) was incomplete, 0.8 Mb of bovine satellite repeats were found in its centromere. The size of the assembled BBUX centromere was comparable to that in sheep (1 Mb) and goats (0.6 Mb). BTAX was submetacentric and its centromere was at a different location from all species examined, with no bovine satellite repeats. Three copies of SATII were detected on BTAX, 3.5 Mb away from the centromere, which could be the remnant of the ancestral centromere.

## Cattle X-Y PAR and non-PAR synteny

The PAR was identified on X-PAR between coordinates 166,757,707 and 173,598,413 in reverse order on the q-arm. The previously published Wagyu Y chromosome[41] PAR was located from position 20,023 to 6,819,277 on the p-arm (Supplementary Fig. 6). The X-Y PAR region contains 31 similar genes to each other aside from the PRP gene (Supplementary Data 11-12).

After the Y-PAR boundary, there are six X-degenerate regions and six ampliconic regions (Fig. 5a, Supplementary Data 13)[41]. There were 13 genes within the X-degenerate regions with X chromosome homologs (Supplementary Data 14). Four genes were in the ampliconic regions, and none of them were found in BTAX, as expected. The

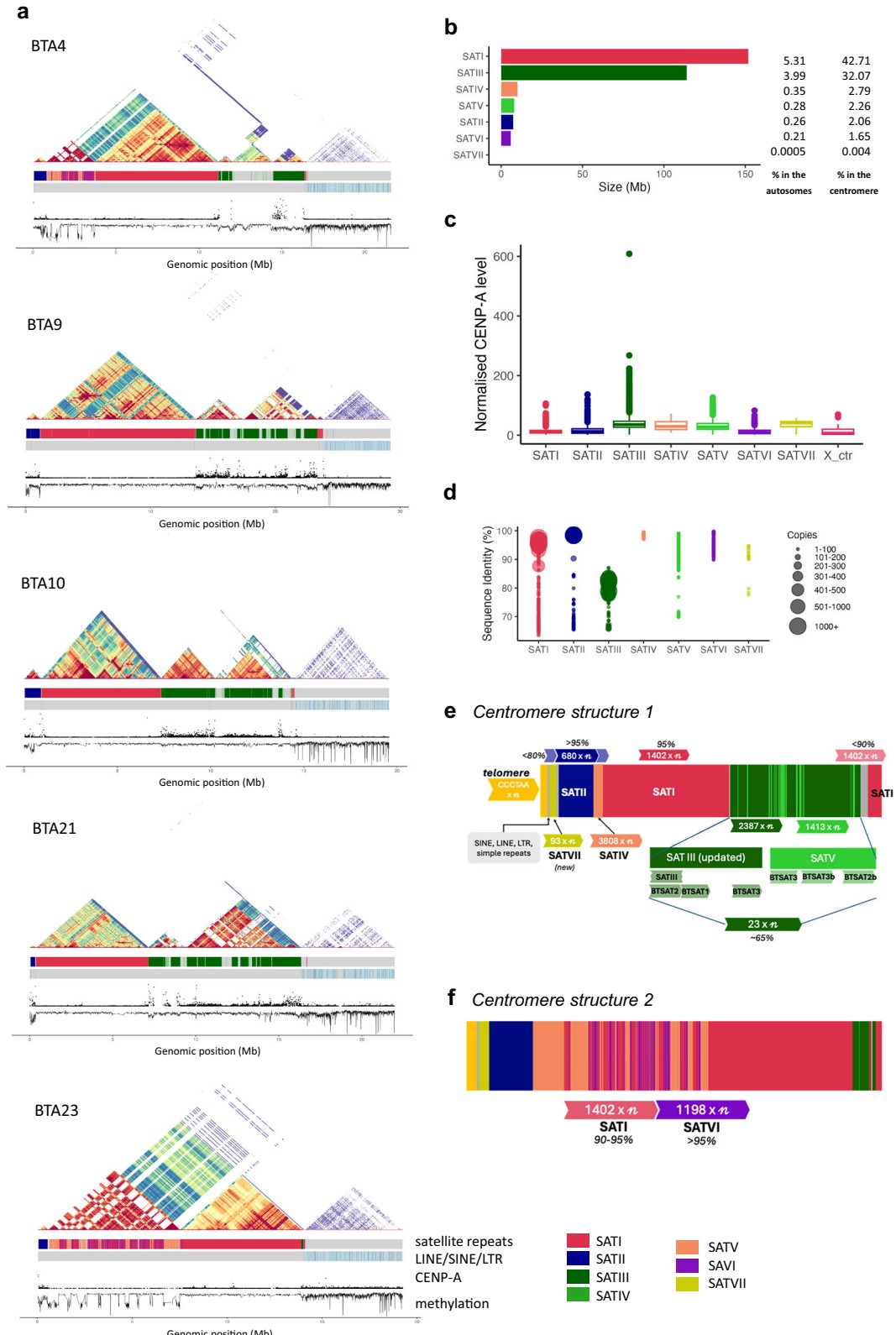

pairwise alignment plot of the X-Y chromosomes showed the two chromosomes were aligned in certain non-PAR regions (Supplementary Fig. 7; Supplementary Note 11).

### Conserved PAR genes and expression in testes

In contrast to human and other primates, cattle have the shortest X-PAR and it is located on the q-arm (Fig. 5b). The cattle X-PAR has unique *BDA2O* and *OBP* genes not found in human and primates. Conversely, cattle do not have *ARSF*, *VCX*, *FAM9A*, and *FAM9B* genes found in primates. The 30 conserved genes among the eight primate species share the same order. Unlike the X-PAR, the Y-PAR in cattle is the longest among the eight species (Supplementary Data 12). The Y-PAR boundary of cattle is *GPR143*, whereas for other species, it is

**Fig. 2 | Overview of the centromere and satellite repeats in UOA_Wagyu_1 autosomes. a** The repeat organization, CENP-A binding and methylation signal of five autosomes with resolved centromeres. BTA4, BTA9, BTA10, BTA21, and BTA23 refer to *Bos taurus* chromosomes 4, 9, 10, 21, and 23, respectively. The heatmap (pyramid) showed sequence similarity, with blue to red indicating increasing similarity. CENP-A signal was high in the bovine satellite repeat SATIII regions, whereas methylation data showed stable signal throughout the centromere region except BTA23, which had lower methylation in SATIV. **b** The total size and percentage of the bovine satellites in centromeres and autosomes. **c** Normalised CENP-A signal of each satellite repeat units and the X centromere. Boxplots show the median (centre line) CENP-A signal, interquartile range (box; 25th and 75th percentiles) and the outliers (points) (Supplementary Data 6). **d** The sequence identity

percentage of the seven bovine satellite repeats and their number of copies. **e, f** The two general structures of the cattle centromere, illustrating the landscape and position of each satellite repeats. The numbers indicate the length of each repeat unit, e.g., "680 x *n*", where 680 represents the length and *n* denotes number of copies. The SATIII consists of the BTSAT2, BTSAT1 and BTSAT3 of the RepBase satellite repeats, and SATV consists of a fragment of BTSAT3, BTSAT3b and BTSAT2b. Both SATIII and SATV consists of diverse 23 bp monomers. BTA1, BTA3, BTA5, BTA8, BTA9, BT10, BTA17 and BTA21 exhibit structure 1 whereas BTA4, BTA12, BTA16, BTA23 and BTA26 display structure 2 (Supplementary Fig. 2). The rest of the autosomal centromere structure are undetermined. The figure is not shown to scale. Source data are provided as a Source Data file.

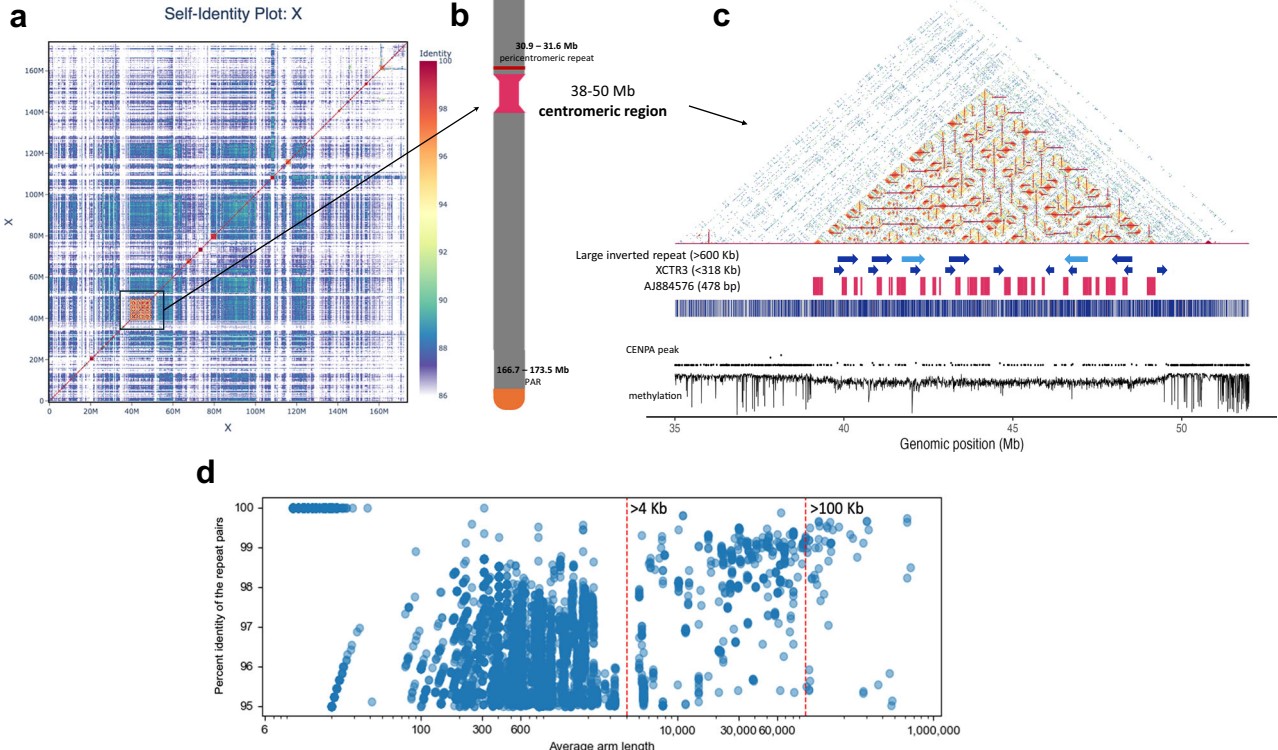

**Fig. 3 | Structure of the cattle X chromosome. a** Moddotplot visualization of the BTAX and the location of its centromere highlighted in boxed outline. **b** An illustration of the cattle X chromosome indicating different key regions. **c** Heatmap of the centromeric region showing the locations of the three specific repeats (Large inverted repeats >600Kb), XCTR3 and the X chromosome specific repeat with

accession number AJ884576, with the CENP-A peak signal and methylation signals. **d** The length and similarity of the inverted repeat pairs in the centromere region. Each point indicates the average pair length. Source data are provided as a Source Data file.

*CD99* or *XG*. Human and Bonobo have a secondary PAR with extra genes that are not found in the other species.

A total of 17,007 genes have been identified as being expressed in cattle testes (Supplementary Data 15). For cattle-specific genes, two copies of *OBP* genes (ENSBTAG00085026078 and ENSBTAG00085025782) were expressed in the testes with a slightly higher level in immature cattle than mature cattle (Supplementary Data 16, Supplementary Fig. 8a). No expression was detected in testes for the *BDA20* gene, but one *BDA20* gene (ENSBTAG00000016820) was highly expressed in ear and skin in mature cattle (Supplementary Fig. 8b).

Eight genes (*PLCXD1, GTPBP6, PPP2R3B, SLC25A6, ASMTL, AKAP17A, DHRSX, CD99*) exhibit conserved expression in testes across five species in the Y-PAR, and they also have a similar expression pattern on the X-PAR (Supplementary Data 16-17). It is noteworthy that *TBL1Y* is also consistently expressed in all the species on the Y chromosome; however, only cattle have it in the PAR. The newly identified gene, *PRP*, which was only found in the cattle Y-PAR, was expressed in testes.

## Discovery of new cattle genes

As UOA_Wagyu_1_Y represents the most complete cattle genome to date, we compared its annotation against the current reference to determine the number and types of newly annotated features. We found 10,566 new features that had not been annotated previously across cattle chromosomes 1–29 and X. Most of these features (95%) were long non-coding RNAs (7985), rRNAs (1340) and protein coding genes (738) (Supplementary Data 18). BTAX had the greatest number of new protein-coding annotations, with 337. However, ten of these new features were genes from the PAR of the X chromosome that had been mis-assembled in ARS-UCD2.0.

There were 37 protein-coding genes within the BTAX centromere, with 24 of them being newly identified i.e. not found in ARS-UCD2.0 (Supplementary Data 19). Gene ontology (GO) term and pathway enrichment analysis of these 37 genes revealed that the BTAX centromere genes are enriched in the "insulin regulation of blood glucose" pathway (Supplementary Fig. 9a, Supplementary Note 3). These 37

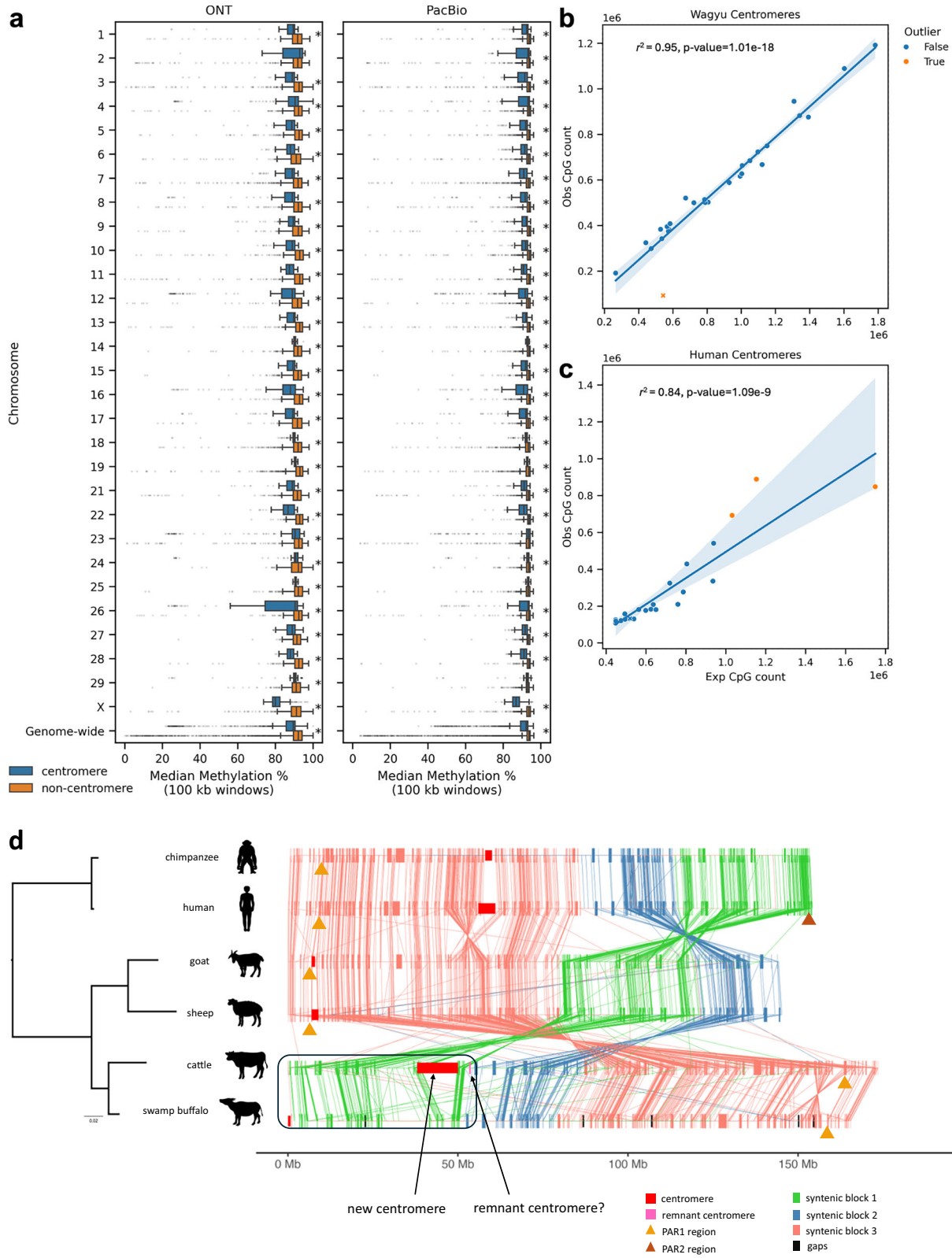

BTAX centromere protein-coding genes were all expressed in the cattle testes (Supplementary Data 20).

## Wagyu-specific genome improved mapping rate and variant calling from short-reads

As genomes become more complete, the mapping rate of sequencing reads tends to improve. To assess this, we investigated the mapping rates of ~10x long-reads and ~11-20x short-reads generated from Wagyu animals. Using long reads from the 20 Wagyu samples in this study, we observed an average mapping rate of 86.84% when mapped to ARS-UCD2.0 (Supplementary Data 21). In contrast, we observed an average mapping rate of 94.80% when aligning to UOA_Wagyu_1_Y. Short-read WGS mapping rates from 11 publicly available Wagyu samples were good for both ARS-UCD2.0 (99.64%) and UOA_Wagyu_1_Y (99.75%) assemblies (Supplementary Data 22). There were 15,968,955 and 16,125,281 single nucleotide polymorphisms (SNPs)

**Fig. 4 | Centromere and CpG methylation. a** Median methylation of each chromosome divided into centromeric and non-centromeric windows at 100 kb and step size of 50 kb. Each point on these plots represents the median methylation of a 100 kb window along the chromosome. The blue box and whisker plots denote centromere methylation and the orange box and whisker plots denote non-centromere methylation. The left panel illustrates the methylation values obtained using Oxford Nanopore Technology (ONT) data and the right panel shows the methylation values obtained using Pacific Bioscience (PacBio) data. Statistically significant results by Mann-Whitney U-test are marked by '*' (Supplementary Data 8). Box and whisker plots follow the standard format, where the left whisker denotes the lower extreme, the left side of the "box" denotes the lower quartile, the vertical line within the "box" is the median, the right side of the "box" is the upper quartile and the right most whisker is the upper extreme. Any points either side of the whiskers are outliers (+/− 1.5 times the interquartile range) **b** Linear regression showing the relationship between the observed CpG count and the expected CpG count in UOA_Wagyu_1 centromeres. Confidence intervals represent the 95% confidence interval. Non-outliers are coloured blue, while outliers are coloured orange; the X chromosome is denoted by an "x". **c** Same as (**b**) but using the centromeres from the human T2T (telomere-to-telomere) genome (CHM13v2.0/hs1). **d** Comparison of the X chromosome showing conserved genes, syntenic blocks, pseudoautosomal region (PAR) and centromere region across different species. The centromere regions are boxed in red, the PAR region is marked with a triangle, and the remnant centromere is shown in pink. The swamp buffalo is not T2T and the gaps are indicated as vertical black bars. Source data are provided as a Source Data file.

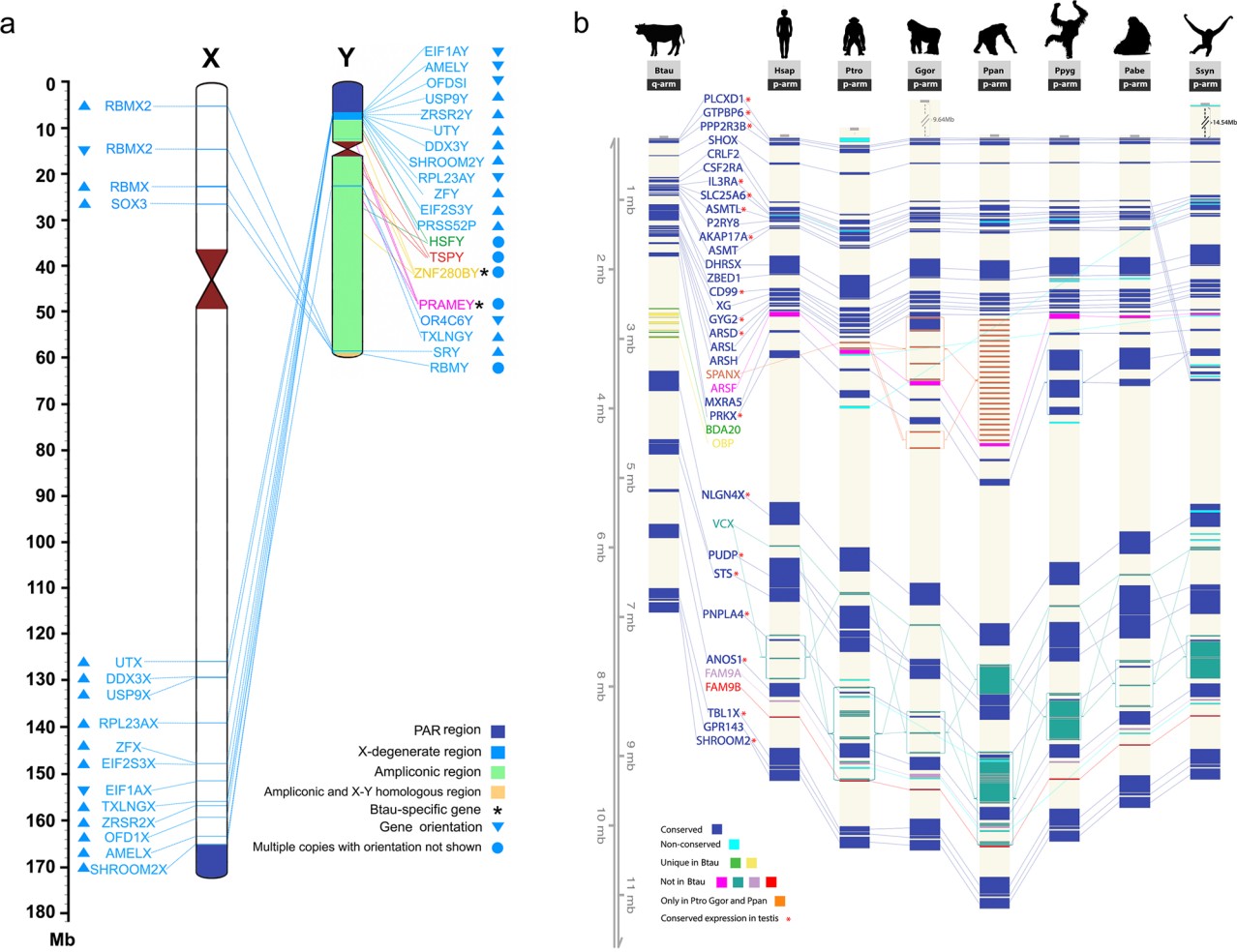

**Fig. 5 | Gene conservation on cattle X and Y chromosomes. a** X-Y homologous genes in cattle non-pseudoautosomal region (non-PAR). There are 102 copies of *TSPY*, 40 copies of *HSFY*, 33 copies of *PRAMEY*, 22 copies of *ZNF280BY* and 11 copies of *RBMY* found in the ampliconic region of cattle Y chromosome. Gene *UTX* is also known as *KDM6A*. Genes in light blue are located in the X-degenerate region, and the genes in other colours are located in the ampliconic region. **b** Comparison of the X-PAR genes in cattle, human and apes. The conserved PAR genes are highlighted in dark blue, and non-conserved PAR genes are highlighted in light blue. Unique gene families, *OBP* and *BDA20*, in cattle are highlighted in yellow and green. Homologous genes are linked by lines across species. Eighteen genes have conserved gene expression in testes across Btau *(Bos taurus)*, Hsap *(Homo sapiens)*, Ptro *(Pan troglodytes)*, Ggor *(Gorilla gorilla)* and Ppyg *(Pongo pygmaeus)*. Source data are provided as a Source Data file.

called using UOA_Wagyu_1_Y and ARS-UCD2.0 assemblies, respectively (Supplementary Data 23).

## Generating and characterising a catalogue of Wagyu SVs
The MCG method works by first selecting the individual that is the most genetically representative of the population e.g. a commonly used sire. It then selects the next sample that captures the most

genetic variation not observed in the first sample, with this process repeated to capture more genetic diversity with less samples[48]. We employed the MCG method to reduce the number of samples needed to capture the most genetic variation within our research budget. The MCG method for selecting 20 animals for resequencing outperformed random sampling in capturing the maximum amount of genetic variation within 20 samples (Fig. 6a; Supplementary Note 12). Four SV

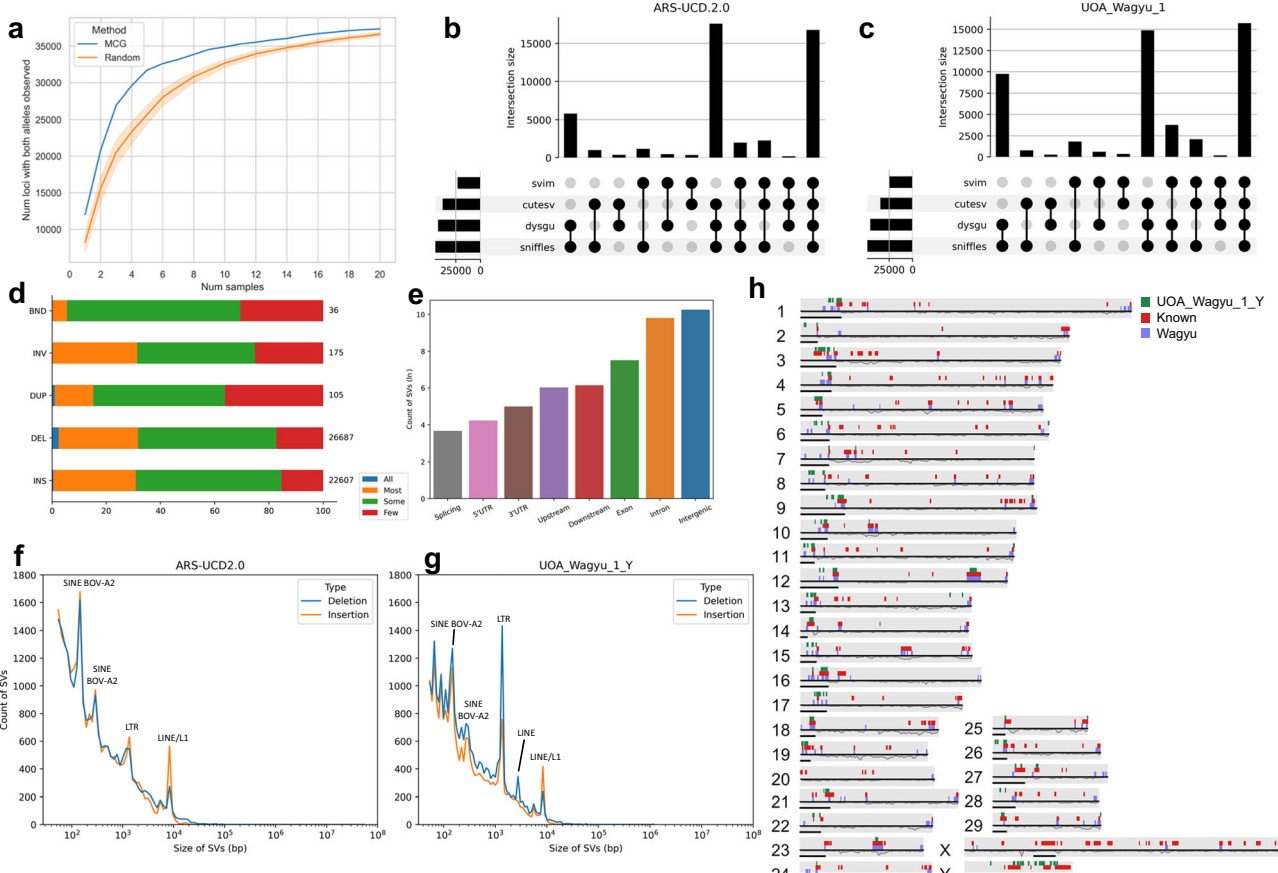

**Fig. 6 | Structural variant (SV) catalogue of Wagyu. a** Line plot showing higher number of both alleles observed when using the MCG method compared to randomly selected animals. Error bars represent the 95% confidence interval. The blue line denotes the MCG method while the orange line denotes random sampling. **b**, **c** Upset plot showing the number of SVs identified by different combinations of SV callers when using ARS-UCD2.0 or UOA_Wagyu_1_Y as the reference. **d** Stacked bar plot showing what proportion of each of the five SV categories are found in all (blue), most (orange), some (green) or few (red) samples. **e** Count of SVs by different type of genomic regions. **f** Kernal Density Estimate plot of the length distribution of insertions and deletions above 50 bp when mapped to ARS-

UCD2.0. The Y-axis is the SV count and the X-axis is the length of the SVs following a log transformation. Deletions are coloured blue and insertions are coloured orange. **g** Same as (**f**) but mapped to UOA_Wagyu_1_Y. **h** Ideogram of UOA_Wagyu_1_Y showing known hotspots (red), hotspots identified based on the high-confidence SVs generated from the 20 Wagyu samples (lilac) and hotspots from SVs identified in UOA_Wagyu_1_Y but none of the other assemblies (green). The top half of each ideogram displays the SV hotspots while the bottom half shows the protein-coding gene density. The horizontal black bar indicates centromere position. Source data are provided as a Source Data file.

callers were used to identify SVs among the 20 samples to minimise the likelihood of false positives. After filtering and merging SVs, we identified 49,610 SVs using UOA_Wagyu_1_Y as the reference and 47,213 SVs when using ARS-UCD2.0 (Fig. 6b, c). We found that regardless of reference genome used, most SVs were insertions and deletions (Supplementary Data 24; Supplementary Note 13). The remaining SVs in descending order of frequency were inversions, duplications and breakends. Regardless of the genome used, the majority of SVs were found in at least three samples (Fig. 6d). The majority of SVs occurred within the intergenic regions of UOA_Wagyu_1_Y (28,437) (Supplementary Data 25). This was followed by SVs occurring in the intronic regions (18,242) and exonic regions (1816). There were 415 and 468 SVs that overlapped with the 1-kb region up or downstream of a TSS, respectively (Fig. 6e). The three prime UTR, five prime UTR and splicing junctions each had progressively fewer SVs overlapping them (148, 69 and 39, respectively) (Fig. 6e).

Consistent with previous studies[49], most observed SVs were relatively short, with two peaks below 300 bp (~145 bp and 285 bp) (Fig. 6f, g). These short insertions were mainly annotated to BOV-A2 (SINEs), which has been observed previously[49]. We detected another insertion peak at around ~1300 bp, which was mainly annotated to long-terminal repeats (LTRs). There was a deletion peak around

~2700 bp detected in UOA_Wagyu_1_Y but not ARS-UCD2.0, which corresponds to LINEs. Finally, there was a peak at ~8500 bp in the ARS-UCD2.0 and UOA_Wagyu_1_Y assemblies that corresponds to LINE/L1 elements.

Regions of the genome that may be Wagyu-specific were examined by constructing a pangenome graph of 13 beef cattle breeds and identifying presence-absence variants (PAVs) among the breeds. From the pangenome graph, anchored to UOA_Wagyu_1_Y coordinates, 283,348 PAVs ("graph PAVs"), each more than 50 bp, were identified. A subset of these appeared Wagyu-specific including 5840 PAVs ("WAG assembly PAVs") unique to UOA_Wagyu_1_Y. There were 133 non-redundant Wagyu-specific hotspots identified from the SVs and PAVs (Supplementary Note 14) (Fig. 6h). These hotspots were then overlayed with protein-coding genes to reveal 509 genes that overlapped a hotspot by at least 50% of their length (Supplementary Data 26); most of these genes were associated with olfactory transduction (Supplementary Fig. 9b).

## Discussion

Our complete cattle X chromosome combined with the recently published cattle Y chromosome[41] and complete sex chromosomes of human[8,37], apes[38], sheep[39] and goat[40] have enabled us to study the

structure and evolution of sex chromosomes in unprecedented detail, including the discovery of a natural neocentromere exclusively found in cattle. The origin of the X chromosome is likely from an ancestral autosome[50], based on the prediction that common centromeric satellite repeats will usually be found in autosomes and sex chromosomes[38,40]. While this prediction holds true for human, apes, sheep, water buffalo and goats, BTAX is surprisingly devoid of shared satellite repeats with autosomes[51]. We detected only three tandem copies of SATII and sporadic presence of SATVI and fragments of SATIV; too few to constitute the megabases of bovine satellite sequences typical of autosomal centromeres. The BTAX centromeric region spans 12 Mb, which is ~5 times larger than the Y centromere[41]. Unlike the HSAX centromere and the BTAY centromere, which consist of highly similar monomeric repeat units that form higher order repeats (HORs) and are tandemly repeated, the BTAX centromere was mainly composed of highly identical inverted repeats with five X centromere specific repeats and contained a higher number of TEs than the non-centromeric regions of BTAX.

Analysis of Cetartiodactyla species showed the general structure of X chromosomes and centromere positioning in pigs, alpacas, and whales are conserved whereas major centromere repositioning occurred in ruminants[52]. Previous cytogenetic work of cattle, buffalo and sheep have suggested transposition of X centromeres, but there has been a lack of clarity on X centromere evolution[52,53]. Our comparative genomics analyses of the synteny, repeats and centromere position across human, apes and livestock revealed the BTAX centromere is a natural neocentromere because its location is not shared with other species, it has low methylation signal, and is devoid of bovine satellite repeats. The low methylation signal supports this region being a new centromere because of the high number of remnant TE content observed. TE insertions are often methylated to inhibit their transcription[54] but subsequent spontaneous deamination will result in low methylation. Spontaneous deamination of CpGs to TpGs is the most common mutation in the mammalian genome[55,56]. As we observed high TE content within the BTAX centromere and a depletion of CpGs compared to TpGs, we hypothesise that the BTAX neocentromere originally arose via expansion of TEs, which were subsequently methylated to repress their transcriptional activity. The TE methylation was then followed by a wave of spontaneous and enzymatic deamination events, leading to an enrichment of TpG dinucleotides and low methylation within the BTAX centromere. This enzymatic deamination may have been mediated by the AID/APOBEC polynucleotide cytidine family of deaminases[57].

In the literature for mammals, we could only find eight species with cases of ENC[30–35,58]. These ENCs are generally associated with CENP-A and therefore, our finding of BTAX neocentromere low CENP-A is unprecedented. CENP-A in cattle is strongly associated with SatIII. The centromere of BTAX lacks any bovine satellites in it and as such CENP-A may only be loosely bound to the centromeric inverted repeats. Future experiment with anti-CENP-A antibody staining will confirm the active X neocentromere. We observed low methylation signal in the cattle X centromere, but other mammalian ENCs have no methylation data on their neocentromeres for a comparison. The large, inverted repeat feature of cattle X neocentromere is not shared by other mammalian ENCs.

Analysis of five cattle autosomes with resolved centromeres showed their centromeric regions consisted of large tandem arrays of seven satellite repeats, which corresponds to the general characteristic of a centromere[19]. However, since we lack T2T for the other chromosomes, some structures may be missed. Nonetheless, based on our work, we found two structural patterns of bovine centromeric region characterised by the presence of SATI and SAVI interspersed repeats. The different repeat sequence identity of the bovine satellite repeats suggests that they are undergoing different homogenization stages[51]. Additionally, we observed a high CENP-A enrichment level in SATIII,

suggesting centromere activity and function within this region. Bias for AT-rich satellite repeats has been observed in other animals[19], but we observed a higher GC content in the identified cattle satellite repeats, which contributed to higher overall GC content in the chromosomes.

The Wagyu genome presented here is not T2T for all chromosomes likely because of insufficient coverage of ultra-long reads (only 18.3x of >100 kb ONT), and the high sequence similarity of cattle centromeric repeats and rDNA has led to unresolved tangles in the genome graphs. These constraints make it harder to completely resolve the cattle genome despite numerous attempts in genome assembly including the use of HERRO corrected reads to boost sequence coverage of HiFi-like reads. Future efforts to complete cattle genome should be directed at increasing ultra-long reads and manually disentangling genome graph, assuming that few unresolved tangles remain post assembly.

Reference genome bias is known to impact variant calling from both genomic and epigenomic data and has been demonstrated in diverse species, such as humans[59–62], cattle[3,63], water buffalo[9] and mice[64]. Having a breed-specific reference genome improved the mapping rate and variant calling of both short and long-reads. We observed a greater number of SVs in Wagyu samples using the more complete UOA_Wagyu_1_Y genome compared to ARS-UCD2.0, highlighting the importance of using a representative genome for the population being studied. This is similar to human T2T CN1 genome that enabled the characterisation of 6,641 more SVs for East Asian samples than when using CHM13 as the reference[36]. Pathway analysis revealed olfactory transduction related genes as being associated with SV hotspots, which is interesting given that olfactory transduction associated genes play an important role in adipocyte differentiation[65–67]. However, further work is needed to clearly identify the genes involved and validate any potential relationship.

The conserved expression in testes of 18 X-PAR genes suggests that these genes play potential roles in spermatogenesis and other aspects of male fertility. Interestingly, all 37 centromere genes, of which 15 are ribosomal proteins, are expressed in cattle testes. These genes are located in a region with an abundance of inverted repeats and this class of repeats is associated with testes expression in mouse and human[68]. The presence of highly homologous large inverted repeats could have facilitated gene conversion to maintain sequence integrity of genes with a role in male fertility. The X-PAR has 293 SVs and ~25,000 SNPs from the 20 Wagyu animals, and it contained one SV hotspot. Unfortunately, to avoid complications with analysis of genetic markers in the non-PAR region of cattle, BTAX has been routinely removed in genomic prediction and genome-wide association studies, despite its importance in milk production and fertility traits[17]. Given the role of the X chromosome in reproduction and the potential importance of its genetic variation, molecular evolutionary analyses and the cattle industry will benefit from its completion.

We present the most contiguous cattle genome to date which consists of a complete cattle X chromosome and four T2T autosomes, characterizing the complex and repetitive regions in the genome such as the centromere, telomere and rDNA. Comparative T2T chromosome analysis unveiled a natural X neocentromere in cattle and this provides the opportunity to investigate centromere formation and evolution in mammals. We produced a high-confidence catalogue of cattle SVs that we anticipate will be useful for global efforts in investigation of the incorporation of this important class of variant in future genome wide association studies and genomic prediction work to select for agriculturally important traits.

## Methods
### Sequencing of the trio (Tuli x Wagyu)
This study was conducted in accordance with the Australian Code for the Care and Use of Animals for scientific purposes with approval number S-2021-049. A hybrid F₁ of a Wagyu dam and Tuli sire was selected for a

trio-binning approach of a haplotype-resolved genome assembly (Supplementary Methods). To haplotype phase the F1 hybrid, blood was collected from the Wagyu dam and semen was collected for the Tuli sire and were sequenced with Illumina short reads to 38x and 40x coverages, respectively (Supplementary Data 27). High quality genomic DNA from the blood samples of the male $F_1$ hybrid was sequenced with PacBio HiFi to 58.1x and ONT to 228.8x (18.3x of these are ultralong defined as at least 100 kb). The HiFi reads were generated from DeepConsensus (SMRT Link v12.0.0.1)[69] and they were screened for adapters with HiFiAdapterFilt (v2.0.1)[70]. About 72.8x of ONT reads were generated with R9.4.1 sequencing chemistry, with the remaining being generated with R10.4.1. Additionally, we generated 421 million 2 × 150 paired end HiC reads and 81.8x of Illumina short reads for the $F_1$. A subset of ONT long reads (~57x coverage, R10.4.1) was corrected using HERRO (Haplotype-aware ERRor cOrrection)[71]. Prior to correction, the reads were filtered to retain only those with a minimum length of 10 kb and a minimum average read quality of QV10. The HERRO-corrected ONT reads were used as additional high-accuracy "HiFi" input because of its improved base-level accuracy.

## Genome assembly and polishing

Genome assembly was performed using Verkko with -C 0 option with PacBio HiFi reads, ONT filtered reads, HERRO corrected ONT reads and parent hapmers (Supplementary Methods). Verkko includes a scaffolding step that incorporates ONT reads to connect and bridge contigs. Further scaffolding was done using YaHS (v1.2a.2)[72] with HiC short reads on the draft assembly. Although, scaffolding with Hi-C data using YaHS software was attempted, two chromosomes (BTA2 and BTA11) were erroneously joined together, hence this was excluded due to misassembly. The draft assembly was aligned with ARS-UCD2.0 to conform orientation with the established cytogenetics for cattle and was checked for misassembly with dotplot. The assembly was polished by aligning PacBio HiFi reads to the draft assembly using minimap2[73] and substituting in homozygous alternate variants from DeepVariant[74] using BCFtools (v1.17)[75].

The final assembly is available at the National Center for Biotechnology Information (NCBI) under the accession number GCA_040286185.1 (UOA_Wagyu_1). For analyses that included the Y chromosome, we concatenated the previously published Y chromosome[41] from ARS-UCD2.0 to UOA_Wagyu_1, herein referred to as UOA_Wagyu_1_Y. The Y chromosome in ARS-UCD2.0 was derived from another, unrelated Wagyu animal[41]. The Tuli assembly was not analysed in this work.

## Assembly evaluation

The assemblies generated from the different runs of Verkko were evaluated by the number of T2T chromosomes in the Wagyu haplotype, presence of telomere[76], N50[77], Merqury QV[78] and switch-error (Supplementary Methods). A custom script was used to generate a table of above-mentioned assembly statistics and to identify contigs/scaffolds that were homologous to ARS-UCD2.0 chromosomes (https://github.com/plnspineda/assembly_initialqc). Genome size and heterozygosity score were estimated with GenomeScope2[79] and meryl (v1.3)[78] k-mer counts generated from Illumina short-reads of the F1. Assembly completeness was computed using compleasm (v0.2.6)[80] and k-mer completeness was computed using Merqury (v1.3).

## Annotation and mining for new genes

The Y chromosome NCBI annotation of ARS-UCD2.0 was combined with Ensembl UOA_Wagyu_1 to create a full annotation for UOA_Wagyu_1_Y (Supplementary Note 3). To identify previously unseen genes, i.e. genes from regions not previously annotated in ARS-UCD2.0, we used Liftoff (v1.6.3)[81] to map the ARS-UCD2.0 annotation to UOA_Wagyu_1 (Supplementary Methods). With the ARS-UCD and UOA_Wagyu_1 annotations on the same coordinate system, we then identified the

features in both annotations that overlapped by at least 90% of their length. Following this, we took the complement of these features to identify new annotations in UOA_Wagyu_1. Six previously unannotated X chromosome centromeric genes did not have Human Genome Organisation (HUGO) Gene Nomenclature Committee (HGNC) names. Using the protein sequences of these genes as input, Basic Local Alignment Search Tool (BLAST)p[82] was performed against the human non-redundant protein sequences (nr; organism is *Homo sapiens*) with default parameters to identify homologous genes (Supplementary Data 19). Annotation files of the UOA_Wagyu_1 are available through Ensembl, accessible at https://ftp.ebi.ac.uk/pub/ensemblorganisms/Bos_taurus/GCA_040286185.1/ensembl/geneset/2024_11/. Alternatively, the annotation can be viewed at Ensembl Genome Browser https://beta.ensembl.org/species/6d661feb-0bc2-4b4f-baa4-9e1e7a0c764f.

## rDNA copies

The number of rDNA copies was determined by aligning 18S, 5.8S and 28S sequences from a representative rDNA sequence (Genbank: DQ222453.1) using BLASTn from BLAST+ (v2.14.1)[82] to the UOA_Wagyu_1 reference genome. BLAST results that showed 18S, 5.8S and 28S in the right order with at least 85% identity, 80% query coverage and an e-value of 1e-10 were counted as complete copies (Supplementary Methods). The complete 35 kb rDNA consensus includes the external transcribed spacer (ETS) and IGS.

## Autosomes repeat annotation and centromere identification

Repeat sequences in the UOA_Wagyu_1 assembly were analysed using RepeatMasker (v4.1.5)[83]. Firstly, the repeats were identified by running RepeatMasker using the default RepeatMasker library and RepBase RepeatMasker Edition version 20191026. This was done to identify known bovine satellite repeats and to establish centromere boundaries. Following manual annotation, we defined seven bovine satellites (SATI-SATVI) and a novel 93 bp satellite, which was denoted as SATVII (Supplementary Data 28).

The presence of SATI-SATVII in the autosomes delineates the centromeric region (Supplementary Methods). The centromere boundary was defined by the presence of bovine satellite repeat clusters. We defined these repeat clusters as consecutive 100 kb windows with at least 20 kb of repeat sequence. Given that all cattle autosomes are acrocentric, the start position of the centromeric region was defined as the first occurrence of the satellite repeat downstream from the telomeric region. The end position of the centromere was the last occurrence of the satellite repeat within the repeat cluster (Supplementary Fig. 10).

## CENP-A enrichment analysis

We obtained CENP-A CUT&RUN data from GSE262830 and followed the CENP-A enrichment analysis previously detailed in the Wagyu Y chromosome work[41]. Raw reads were trimmed, aligned to UOA_Wagyu_1_Y, filtered and the resulting BAM files were used to identify CENP-A peaks with SEACR[84]. The CENP-A peaks were overlapped with satellite repeats to identify the CENP-A peaks associated with these repeats (Supplementary Methods). A Kruskal-Wallis test was performed to compare the differences in CENP-A peak distribution across the satellite repeat types.

## Whole X chromosome repeat annotation

We implemented Comprehensive ab initio Repeat Pipeline (CARP)[45], a de novo repeats identification tool, to identify new repeats. Dispersed repeats on the X chromosome were identified using default parameters, which included a sequence identity of 94% and a minimum alignment length of 400 bp. RepeatMasker was utilized to calculate the coverage of the CARP consensus sequences (Supplementary Methods; Supplementary Note 9).

### X chromosome centromeric repeat annotation

The centromeric region in the X chromosome was defined by the ~12 Mb repeat block (X:38,000,000–50,000,000) visualised with ModDotPlot. This repeat block contained the X centromere specific repeat (Accession number: AJ884576)[44]. The BTAX centromere did not display characteristics similar to autosomal centromeres, but rather was composed of large inverted repeats and sporadic repeat sequences. To extensively characterise the repeats within the X centromeric region, we have used different CARP parameters to characterise potential monomeric and inverted repeats (Supplementary Methods).

The inverted repeats within the X centromere were identified with Inverted Repeats Finder (IRF v3.08)[68] using the parameters "2 3 5 95 10 20 2000000 12000000 -t7 12000000 -d -ngs". The paired repeats were filtered for ≥95% sequence identity. The largest repeat arm was aligned against the smaller inverted repeats using BLASTn[82] to identify nested repeats.

### Methylated CpG and CpG enrichment analysis

To determine the methylation signals from PacBio HiFi reads, we mapped the reads to the UOA_Wagyu_1_Y assembly using pbmm2 (https://github.com/PacificBiosciences/pbmm2) (v1.14.99) and then extracted methylation information with pb-CpG-tools (https://github.com/PacificBiosciences/pb-CpG-tools) (v.2.3.2). Similarly, to determine the methylation signal in ONT reads, we mapped the ONT reads to UOA_Wagyu_1_Y using "dorado aligner" (https://github.com/nanoporetech/dorado). We then quantified CpG methylation using modkit (https://github.com/nanoporetech/modkit) (v0.2.8) (Supplementary Methods).

The Mann-Whitney test was used to determine whether the median methylation was different between centromeres and non-centromeres. We repeated this using publicly available methylation data for CHM13v2.0. Jellyfish (v2.3.1)[85] was used to count all dinucleotides that occurred within and without the centromeric regions of UOA_Wagyu_1_Y. A Fisher's exact test was used to determine if the CG dinucleotide was enriched in centromeric regions compared to non-centromeric regions (Supplementary Methods).

### Evolution of X centromere across species

The conservation of syntenic regions on the X chromosome was investigated by comparing gene alignments and orthologous X chromosomal genes of cattle, swamp buffalo, sheep, goat, human and chimpanzee with OrthoFinder (v2.5.5)[86]. The centromeric region in sheep[39], goat[40], human[8] and chimpanzee[38] were determined from the published position of centromere. The centromeric region in swamp buffalo[9] was identified by searching for bovine centromeric satellite repeats using RepeatMasker.

### Conservation of pseudoautosomal regions (PARs)

To identify the PARs, pairwise alignments of X and Y chromosomes were carried out for cattle using similar methods as described in Liu et al.[87] (Supplementary Methods). Known PARs of human and primates including Chimpanzee (*Pan troglodytes*), Gorilla (*Gorilla gorilla*), Bonobo (*Pan paniscus*), Bornean orangutan (*Pongo pygmaeus*), Sumatran orangutan (*Pongo abelii*) and Siamang (*Symphalangus syndactylus*) were extracted according to published coordinates[38] with SAMtools (v1.10)[75]. The PARs from these species were compared with PARs of UOA_Wagyu_1_Y to study conservation of synteny. The human and primate gene annotations were retrieved from NCBI. BLASTp was used to find homologous genes (Supplementary Methods).

### X-Y synteny and homologous genes in cattle

To study the cattle X-Y homologs in the non-PAR, the Wagyu X and Y chromosomes were aligned to one another with minimap2 (v2.28). The resulting PAF alignment file was read and plotted by the readPaf and plotMiro functions from R package SVbyEye (v0.99.0)[88], respectively.

The gene annotation and protein sequences for Wagyu X and Y were collected from EBI and NCBI, respectively. To identify the homologous genes, the protein sequences from Y chromosome genes were aligned to protein sequences from X chromosome genes with BLASTp. The homologous genes were identified based on sequence identity and query coverage, both ≥70%. The exception was for *SOX3*, which is an X-linked gene related to *SRY*[89]. The annotation of X-degenerate regions and campliconic regions on the Y chromosome followed the published Y chromosome[41].

### Testes conserved and cattle specific gene expression

The publicly available testes RNA-seq data of cattle, human, chimpanzee, gorilla and Bornean orangutan were downloaded from NCBI (Supplementary Data 29). Bonobo, Sumatran orangutan and siamang were excluded from gene expression analysis because we could not find at least two samples to represent these species. Detailed RNA-seq analysis is given in Supplementary Methods.

### Mapping short reads to the Wagyu genome

Publicly available Wagyu short reads were downloaded from NCBI and mapped to UOA_Wagyu_1 genome for variant calling using the method described in Angus and Brahman assembly work[3] (Supplementary Data 22).

### Long-read resequencing of 20 Wagyu animals

The MCG method[48,90] was used to identify 20 Wagyu animals that represented the most genetic diversity within the population from which the UOA_Wagyu_1 maternal haplotype was sourced. The purpose was to capture more structural variants than randomly picking animals from the population (Supplementary Methods).

We sequenced 20 pure Wagyu samples chosen by the MCG method using an ONT PromethION 2 Solo (PRO-SEQ002) with the SQK-NBD114-24 barcode kit and R10.4.1 flow cells to ~10x coverage. Sequenced data from each sample was base called using Dorado (https://github.com/nanoporetech/dorado) (v0.6.0) with '--min-qscore 9' and the "SUP" models (Supplementary Methods) these were then saved as unmapped BAM files to preserve the methylation tag information. We used a bed file with the coordinates of each placed chromosome for the respective genome with Qualimap bamqc[91] to determine mapping rate for a given sample and reference genome.

### Detection of high-confidence SVs

We implemented a custom Nextflow[92] pipeline for long-read variant analyses (https://github.com/DaviesCentreInformatics/LR-variantCaller). Briefly, unmapped BAM files had their quality assessed with NanoPlot (v. 1.4.2)[93]. They were then filtered with Filtlong (v. 0.2.1) (https://github.com/rrwick/Filtlong), removing sequences <200 bp in length, and reassessed with NanoPlot. Filtered reads were aligned to ARS-UCD2.0[2] and UOA_Wagyu_1_Y using 'dorado aligner' with default parameters. We identified SNPs using Clair3 (v.1.0.5)[94] and SVs using four callers, Sniffles2 (v2.3.2)[95], cuteSV (v2.1.0)[96], SVIM (v.2.0.0)[97] and DYSGU (v1.6.2)[98] (Supplementary Methods). SVs from these four tools were merged with SURVIVOR (v1.0.7)[99], with only SVs identified in at least two samples by at least two callers being retained for further analysis; these analyses were completed for each reference genome. In situations where multiple SV callers identified an SV but assigned different genotypes within a single sample, we selected the genotype based on the following priority order: Sniffles2 > cuteSV > SVIM. This order was based on recent work that benchmarked multiple SV callers[100]. Finally, we ignored SVs that occurred on the Y chromosome as only one sample was male.

### SV annotation and Wagyu-specific hotspot identification

We annotated the high-confidence SVs using ANNOVAR[101]. To generate a list of "known SVs", we identified SVs among 13 cattle assemblies using a pangenome graph approach (Supplementary Methods). We

constructed the pangenome graphs using the PanGenome Graph Builder (PGGB) (v.0.6.0)[102] and then identified presence-absence variants (PAVs) with odgi (v.0.8.6)[103]. Hotspots were identified using the hotspotter function from the primatR package.

## Data availability

The PacBio HiFi reads, ONT reads, Illumina paired-end reads and Hi-C reads generated in this study have been deposited in the NCBI database SRA under BioProject PRJNA1117663. The genome assembly is available at NCBI database under the accession number GCA_040286185.1 [https://www.ncbi.nlm.nih.gov/datasets/genome/GCA_040286185.1/]. The annotation file of the UOA_Wagyu_1 is available in the Ensembl database [https://ftp.ebi.ac.uk/pub/ensemblorganisms/Bos_taurus/GCA_040286185.1/ensembl/geneset/2024_11/]. The 20 Wagyu samples ONT long reads generated in this study have been deposited in the NCBI database SRA under BioProject PRJNA1164319. The cattle short reads sequences used in this study were publicly available in the NCBI database with accession numbers DRR397986, DRR397987, DRR397988, SRR23375810, SRR23375811, SRR23375814, SRR23375816, SRR23375817, SRR23375818, SRR23375808 and SRR23375809. The CENPA CENP-A CUT&RUN data were publicly available in the GEO database under the accession GSE262830. The testes RNA-seq data of cattle, human, chimpanzee, gorilla and Bornean orangutan were publicly available in the NCBI database under Bioprojects PRJNA565682, PRJNA1072142, PRJEB46995, PRJNA75899, PRJNA556935, PRJNA30709, PRJEB48233, PRJEB48233, PRJNA304995, PRJNA587108, and PRJNA293451. More details of the public RNA-seq data were provided in Supplementary Data 29. Source data are provided as a Source Data file and large files were available in Zenodo (https://doi.org/10.5281/zenodo.17201175). Source data are provided with this paper.

## Code availability

Code used to perform the analyses throughout this manuscript are available in the Supplementary Information, https://github.com/plnspineda/assembly_initialqc https://doi.org/10.5281/zenodo.17197004) and https://github.com/DaviesCentreInformatics/Cattle_T2TX (https://doi.org/10.5281/zenodo.17196896).

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

## Acknowledgements

This work was supported with supercomputing resources provided by the Phoenix HPC service at the University of Adelaide. The work was part funded by the JS Davies bequest to the University of Adelaide. The work was supported in part by funds from USDA-ARS. The use of trade names or commercial products in this manuscript is solely to provide specific information. It does not imply recommendation or endorsement by the US Department of Agriculture. USDA is an equal opportunity provider and employer. We also thank the DOST-SEI Foreign Graduate Scholarship program for financially assisting P.S.P. We thank Joe Grose for artificial insemination of his Wagyu cow with Tuli semen, raising and bleeding the calf. Ensembl receives majority funding from Wellcome Trust [222155/Z/20/Z] with additional funding for specific project components. Ensembl receives further funding from The Biotechnology and Biological Sciences Research Council [BB/W019108/1, BB/T015608/1, BB/X018695/1]; UK Medical Research Council [MR/S000453/1]; Wellcome Trust [226458/Z/22/Z, 226083/Z/22/Z]; the European Molecular Biology Laboratory (EMBL) core funding and the EMBL transversal research themes funding under the new scientific programme. This project has received funding from the Horizon Europe programme under Grant Agreement Number 101094718 (EuroFAANG). Views and opinions expressed are however those of the author(s) only and do not necessarily reflect those of the European Union or the European Research Executive Agency (REA). Neither the European Union nor the granting authority can be held responsible for them.

## Author contributions

C.B., W.S.P., and W.Y.L. conceptualized and designed the study. W.Y.L., T.P.L.S., and B.D.R. designed the sequencing experiments. T.C. conducted the Hi-C and ONT sequencing for 20 animals. P.S.P. and W.Y.L. performed the genome assembly and analyses. P.S.P., L.Z., D.L.A., and C.D. analyzed the repeat sequences, whereas Y.R. conducted the PAR genes and testes expression analysis. C.M. performed the structural variant and gene annotation analyses. P.S.P., Y.R., C.M., and W.Y.L. carried out all other analyses and interpreted the data. J.P.S., L.H., and F.J.M. annotated the genome. P.S.P., C.M., Y.R., W.Y.L., W.S.P., C.B., T.P.L.S., and B.D.R. wrote, reviewed, and edited the manuscript.

## Competing interests

The authors declare no competing interests.
