## [Transparent Peer Review file · Nature Communications]

Insights into natural neocentromere evolution from a cattle T2T X chromosome

Corresponding Author: Dr Wai Low

Version 0:

Reviewer comments:

Reviewer #1

(Remarks to the Author)

The paper by Pineda et al entitled “Cattle T2T X Chromosome: Insights into Natural Neocentromere Evolution” describes a newly assembled cattle genome, particularly focusing on BTAX and its centromere evolution. The work conducted and detailed explanations of both results and methods are commendable, and the conclusions are sound. I have some minor comments on the text that might improve readability. As in all Nature Communication manuscripts, quite a lot of the material is in Supplementary information, and the main text tends to be short. However, in this case the main text might benefit from some context regarding the analyses being done, particularly in the results section. I suggest adding one sentence at the beginning of each section in results with a bit of detail of what is being analysed, for example:

1. In the section “Genome assembly”, please describe the animal used. It starts by mentioning various versions of assemblies without any context.
2. In the section “Wagyu-specific genome improved mapping rate and variant calling from short-reads”, please add some details on the number of samples used and the reasoning of the analysis. The section starts directly with coverage stats, but doesn't mention of what. Only in methods there's a description of the samples, making it very cumbersome to understand the meaning of the results.

Other comments:

1. No details of gene annotation pipeline are given, only on the comparison of annotations between ARS-UCD2.0 and UOA_Wagyu_1_Y.
2. Where is the CENP-A raw data from? There are no details in methods nor Supplementary information/methods
3. Line 175 – What does CDR stand for?
4. Figure 6 is really interesting, comparing the different algorithms to detect SVs, but it's barely mentioned in the main text nor in the Supplementary information. Could the authors maybe expand this analysis? Do you see a pattern between those SVs called for all algorithms to those unique to one/two? And which ones are the ones then analysed in the rest of the panels of this figure?
5. For the BTAX evolution discussion, it might be worth looking into <https://www.mdpi.com/2073-4425/8/9/216>
6. There are track changes and comments still on the Supplementary Methods section, with some good points that should be addressed.

(Remarks on code availability)

Not all code is available, I could only review the github repository to quality control the assembly. Some lines of code to run several tools are provided in the Supplementary Methods, and they work fine.

Reviewer #2

(Remarks to the Author)

The paper by Pineda et al. describes a haplotype-resolved assembly of a Tuli × Wagyu F1 hybrid cow, constructed using multiple sequencing technologies (PacBio HiFi, ONT—including ultra-long reads—Hi-C, and Illumina). Illumina reads were also generated for both parents to support the phasing process. The genome assembly comprises five chromosomes with telomere-to-telomere (T2T) status (i.e., chromosome-scale contigs with a resolved centromere and telomeric repeats at both

ends), including the X chromosome. In addition, it includes 15 other chromosome-scale contigs that do not meet the T2T criteria, along with other chromosomal contigs and a total of 17 gaps. This assembly enables, for the first time, a detailed description of the structure of the X chromosome—specifically, the position and composition of its centromere. Based on orthologous positions of neighboring genes in related genomes and sequence composition analysis of the predicted centromere, the authors conclude that it represents a neocentromere that emerged in the ruminant lineage. The authors also demonstrate how this assembly makes it possible to describe the composition and structure of autosomal centromeres in cattle, to clarify the nature of the conservation of the pseudoautosomal region between cattle and primates, to expand the cattle gene catalog, and to characterize the structural variations in the Wagyu breed.

Overall, the results presented here represent a significant amount of work using state-of-the-art sequencing technologies and methodologies. However, both the work itself and, perhaps more importantly, the way the results are presented raise some major concerns, which we detail below.

Major concerns:

1/ X chromosome neocentromere: This work identifies, for the first time in cattle, the location of the centromere of the X chromosome. More strikingly, the authors indicate that it is a neocentromere, the centromere is not found at the expected location given the orthology map with related species (water buffalo, sheep, goat, human and chimpanzee). This observation enables the authors to further define the specific architectural features of the bovine X chromosome centromere which are summarized in the abstract (low CENP-signal, low methylation, among others) in contrast to the autosomal centromeres. This raises a number of questions.

The authors report a neocentromere on the cattle X chromosome, identified based on sequence features (cluster of inverted repeats) but lacking a CENP-A signal. To the best of our knowledge, there is no example in the literature of (neo)centromeres lacking an epigenetic signal such as CENP-A binding. This is supported by ref 10 (first key point), ref 18 in the manuscript, and ref 40 “From fission yeast to humans, the histone H3 variant CENP-A was demonstrated to be the epigenetic mark for centromere identity and function”. Since this result is central to the paper, the introduction should present in a more structured way current knowledge on neocentromere formation. Citing ref 20 (line 63) the authors mention “a different type of neocentromere (ENC)... displaying few of the centromere landmarks” suggesting that the cattle X-chromosome neocentromere described in the manuscript belongs to this class. But ref 20 precisely describes a neocentromere that lacks specific sequence features (centromeric satellites) but is associated to a CENP-A signal.

We are not disputing the result reported by the authors, but the way it is presented is unsatisfactory, and as a result, the arguments put forward to support this thesis do not seem entirely convincing.

2/ T2T assembly: The quality of the assembly presented in the paper raises a number of issues.

The current assembly achieves a T2T status only for 5 out of 30 chromosomes, in contrast to the previously published T2T assemblies of two ruminant species (ref 31 and 32 for sheep and goat respectively), for which a T2T status was obtained for all chromosomes. The authors do not discuss the reason that might explain the differences. The reason is most probably related to the limited sequencing depth of ultralong reads (18.3X) in sharp contrast to the assembly protocol for the sheep and goat T2T assemblies (189X and 114X of ultralong ONT reads for sheep and goat respectively). This is probably the motivation for testing various combinations of input data, parameters and software for the assembly process. But the comparison with other approaches is never discussed (mentioned only briefly, lines 78-79 main manuscript “to contrast strengths/weaknesses of previously described approaches”) nor is the rationale for testing the combinations. A discussion about the failure to assemble all chromosomes at the T2T stage is, at the very least, necessary in our view.

In the Genome assembly and polishing section of the Methodology, it is explained that, when scaffolding with HiC data, no contigs were scaffolded correctly. This is difficult to understand without additional explanation, and this should be at least discussed.

In the “Wagyu-specific genome improved mapping...” of the Results, it is explained that the mapping rate to the UOA_Wagyu_1_Y was lower than to the ARS-UCD2.0, but that this mapping rate becomes higher after removing centromeric sequences. It is not reasonable to think that, for a given assembly, some regions should be carefully masked before performing mapping (masking a genome is not recommended for mapping

<https://github.com/lh3/minimap2/issues/654>). This low mapping rate of the T2T assembly contrasts with the increased mapping rate observed for the goat (ref 32 Supplementary Fig. 10) and a reduced error rate for sheep (ref 31 Fig 4a)

Overall, whether regarding the assembly process or the elements supporting the neocentromere hypothesis, the relevant information is very difficult to locate in the paper. This brings us to the main criticism of the paper.

3/ Manuscript organization: The way the manuscript is structured poses a major problem for the reader. Information related to the results and methods appears in different sections with no apparent logic. We give an example to illustrate this point. For the assembly process, the “various combinations of input sequence data...” are first mentioned in the Results section (line 78), but those combinations are only understandable by gathering information from the Methodology section, the Supplementary notes, Supplementary Methods and the Supplementary Table 1. The reader has to make his own calculation to understand how the 228.8X of ONT data are used (72.8X of R9.4.1 (line 426, Main Manuscript) 57X of R10.4.1 \geq 10kb (line 429 MM and line 401 Supplementary Methods), 99X of R10.4.1 \geq 30kb (line 404, SM)). We note that sequencing depth information for the sheep and goat T2T assemblies (ref 31 and 32), for all the technologies, are available in the very first sentence of the results section, which immediately makes it possible to know them.

For all the above reasons, we believe that this manuscript is not yet suitable for publication and requires an extensive rewrite. Below, we provide some examples of problematic points, following the order in which they appear in the manuscript.

Minor concerns:

Introduction

Line 42: ref 8 is not appropriate. The paper deals with telomeres, but nothing is said about the challenge they pose in

assembly

Line 45: ref 10 is not appropriate for the specific problems caused by repetitive sequences for the assembly of allosomes

Line 46 "X chromosome is of particular importance as it is found in all mammalian cells". Are there examples of chromosomes not appearing in all mammalian cells ?

Line 47 Emphasis is made on intelligence, intellectual disability, mental function (with reference 12 appearing twice). For the reader, such emphasis is difficult to understand since this aspect is not mentioned later on in the paper

Lines 53-64 The description is not appropriate. It is mentioned that ENC for "evolutionary centromere" is a different type of neocentromere that displays few of the centromere landmarks, without providing a description of the different types (or the other type) of neocentromeres.

Lines 72-73 The T2T assemblies sheep and goat should be mentioned here (ref 31 and 32 respectively)

Results

Line 84 and line 97. The unplaced scaffolds are mentioned without any reference to the section of the Methods explaining the scaffolding process.

Line 114 "rDNA arrays prevented the complete resolution of BTA4", How this array was detected on BTA4 is not explained. A reference to the corresponding section in the Supplementary Methods should be added, and this section should mention BTA4.

Line 168 A reference to CARP (ref 70) should appear here the first time it is mentioned.

Line 171 The TE described (Bisbis, LTR11B_BT) are most certainly not 849kb and 477kb long respectively, the sentence should probably be rephrased

Line 187 A reference to the Supplementary Methods section where expected CpG count is defined should be provided

Line 264 The gene annotation process is never described, but the reader understands that it was made using the Ensembl pipeline

Line 280 The reduced mapping rate issue has already been mentioned above

Line 310 ref 30 is not related to BOVA2 SINE.

Discussion

Line 389 The proposed relationship between Wagyu SV hotspots targeting olfactory transduction associated genes and breed's superior feed efficiency and marbling is an overstatement that is not supported by the associated references (ref 53 and 54)

Methodology

Line 429 The correction reads using HERRO of a first batch of ONT reads (57 x R10.4.1 with minimum read length of 10kb), appears here for the first time, with no apparent reason

Line 446 ref 24 is inaccurate here but correctly placed line 448

(Remarks on code availability)

Git repository with code associated to the manuscript

Assembly initial QC: https://github.com/plnspineda/assembly_initialqc

The code is not usable for the submitted assemblies without substantial code modification (either X or Y is absent from the submitted haplotypes)

LR-variantcaller pipeline <https://github.com/DaviesCentreInformatics/LR-variantCaller>

"is designed to be run on the University of Adelaide's HPC, Phoenix" as explained in the README.md file. This code is intended for internal use.

Cattle T2T code repository https://github.com/DaviesCentreInformatics/Cattle_T2TX

Contains essentially the R code to produce the figures. While this is very useful, it does not make it possible to understand and replicate the various steps of the analysis. When the code is provided (Testis Gene expression), the very large number of absolute paths highlights that the purpose of sharing it is not to enable readers to reproduce the analyses. And in some cases, it simply refers back to the paper

https://github.com/DaviesCentreInformatics/Cattle_T2TX/tree/main/assembly_evaluation

Reviewer #3

(Remarks to the Author)

This manuscript reports on cattle T2T chromosomes including a T2T X chromosome. Authors use this resource to look into the evolution of centromeres. Methods and approaches appear sound. Overall this appears as a useful manuscript for the bovine genomics community but it is very descriptive and contains a substantial portion of text and analyses that appear off topic.

Authors compare genes annotated on the Wagyu X, Y, and PAR with those annotated in other species. Would be good to clarify early on how the new Wagyu genome was annotated (e.g., in the first Results paragraph), and if the annotation is public. I was unable to find Ensembl entries for the Ensembl IDs mentioned in e.g., Table S15

population genetic analyses presented in lines 278 - 326, and lines 380-391 are off-topic given the title of the manuscript. Moreover, I may have missed it, but couldn't find anything about the short read sequencing and variant calling in methods.

Given the very high transcriptional complexity of testes, I wonder if the fact that genes are expressed in testis indicates indeed that they play a role in fertility (line 34, lines 393ff)

please use consistent terms; e.g., "p arm" vs. "p-arm"

Here are a couple of line-specific comments:

line 23: "The cattle genome is crucial for understanding ruminant biology" - this is a very unspecific sentence. Are you referring to a completely annotated reference sequence?

lines 39-41: "bovine" & "bovids" - the cited references refer to cattle, not bovids.

line 66: "complete gapless" - complete can be deleted

line 77: would be good to present at least some information about the data used for the assembly process, as well as the DNA source (sex and tissue). Otherwise readers need to look that up in the methods.

l. 126: not clear if this was estimated from the HiFi reads, or from an assembly built from HiFi reads

line 175: please define CDR

line 177: is there an "in" missing?

line 227: similar to what? please clarify

line 248: the number of 17k testis-expressed genes is much lower than the 20k reported in <https://doi.org/10.1038/s41467-024-44935-7>

line 267: are the 738 newly identified protein-coding genes functional, i.e., can they be confirmed with e.g., RNAseq?

line 279ff: authors need to provide more details at the beginning of the paragraph what they did. How many samples were considered? average coverage of what?

line 292: same as above. More details needed. What is the "MGC method"? Are these 20 samples the same as used above for the alignment and variant calling? Did you consider only SVs identify by all 6 callers? Kind of odd that most SVs appear to be present in at least 3 samples, wouldn't a large fraction of singletons be expected? did you use any functional data to verify TSS?

line 413: this appears as an overly bold claim which is not substantiated by any results. How will the catalogue of SVs inform future GWAS and genomic prediction?

line 419. Clarify why the F1 is a hybrid. Are the parents from different species?

line 430: ref for Herro needed?

line 442: ref for minimap needed?

there are a couple of author comments left in the Supplementary Notes document (e.g., at page 28, or for Supplementary Figure 4) which the authors should remove

(Remarks on code availability)

Version 1:

Reviewer comments:

Reviewer #1

(Remarks to the Author)

The authors have addressed all my comments. They have expanded the main manuscript giving enough context in each section and included more details on BTAX and CENP-A analyses as I requested.

(Remarks on code availability)

Code now is fully available, but I didn't test it. However, the data is not in the github repo and I'm not sure how reproducible it would be.

Reviewer #2

(Remarks to the Author)

The authors have thoroughly addressed all the comments.

I have only a few minor additional comments.

1. Unless I have overlooked it, the way telomeres are detected in the assembly is not described (line 219 refers to the supplementary information, most certainly to the paragraph Telomere length variations, but how the units are detected is not explained). A description or at least a reference (Pineda et al. 2024 ?) should be provided.

2. Lines 272-276. A reference for BisBis-1.8 (3657bp, Dfam or RepBase ?), LTR11B_BT (1053bp, RepBase ?) and BosInd-1.103 (8366bp, Dfam or RepBase ?) would help clarify the manner in which these TEs span the BTAX centromere.

(Remarks on code availability)

The repository for the code that enables reproduction of all the analyses has been considerably and comprehensively improved. I believe this will be of great value for the readers.

Reviewer #3

(Remarks to the Author)

Authors have addressed most of my comments and made appropriate changes in the manuscript.

However, the population-genetic analyses still appear off-topic and don't contribute much. The title of the manuscript and the abstract (and to a large extent also the introduction) indicate that the manuscript addresses the cattle sex chromosomes and centromere evolution. The SV analyses from 20 long-read sequenced samples aren't mentioned at all. Moreover, these SV analyses don't contribute to the characterisation of centromeres or cattle sex chromosomes. The statement that such analyses are quite common is odd as the references cited (Dai et al., Yang et al.) focus on improved assemblies whereas the current manuscript deals with centromere evolution. I stand by my initial assessment that analyses presented in lines 405-462 are off topic.

(Remarks on code availability)

RESPONSE TO REVIEWERS' COMMENTS

Reviewer #1 (Remarks to the Author):

The paper by Pineda et al entitled "Cattle T2T X Chromosome: Insights into Natural Neocentromere Evolution" describes a newly assembled cattle genome, particularly focusing on BTAX and its centromere evolution. The work conducted and detailed explanations of both results and methods are commendable, and the conclusions are sound.

I have some minor comments on the text that might improve readability. As in all Nature Communication manuscripts, quite a lot of the material is in Supplementary information, and the main text tends to be short. However, in this case the main text might benefit from some context regarding the analyses being done, particularly in the results section. I suggest adding one sentence at the beginning of each section in results with a bit of detail of what is being analysed, for example:

We have ensured more context is given in the main text instead of all information in the Supp Info. In particular, the Results at lines 110-155, 186-190, 210-212, 222-231, 254-255, 281-283, 374-377, 394-400 and 406-413 have been edited for more context.

1. In the section "Genome assembly", please describe the animal used. It starts by mentioning various versions of assemblies without any context.

We appreciate the reviewer's input and have now started this section at lines 110-155 to add context on the animal used with:

"A male calf from a Wagyu dam and Tuli sire cross was selected for haplotype-resolved genome assembly. We generated 228.8x ONT long-read coverage including 18.3x ONT ultra-long (>100 kb), 58.1x PacBio HiFi, ~421 million Proximo Hi-C reads, 81.8x F1 Illumina short-read, and 78x parental Illumina short-read coverage (Supplementary Table 27)."

2. In the section "Wagyu-specific genome improved mapping rate and variant calling from short-reads", please add some details on the number of samples used and the reasoning of the analysis. The section starts directly with coverage stats, but doesn't mention of what. Only in methods there's a description of the samples, making it very cumbersome to understand the meaning of the results.

We appreciate the reviewer's comments and have addressed this section at lines 394-396 to improve clarity. We have begun this section with:

"As genomes become more complete, the mapping rate of sequencing reads tends to improve. To assess this, we investigated the mapping rates of ~10x long-reads and ~11-20x short-reads generated from Wagyu animals. Using long reads from the 20 Wagyu samples in this study, we observed..."

Other comments:

1. No details of gene annotation pipeline are given, only on the comparison of annotations between ARS-UCD2.0 and UOA_Wagyu_1_Y.

We thank the reviewer for bringing this to our attention. We have made reference to this annotation pipeline clearer in the main text and Supplementary Information.

In the main text at lines 186-190:

“The genome annotation was performed by EBI using Ensembl Gene Annotation system, which integrated both short- and long-read transcriptomic data with protein homology model, with emphasis for gene models with transcriptomic data support (Supplementary Information).”

In the Supplementary under section “Gene annotations of UOA_Wagyu_1 and its comparison with ARS-UCD2.0 and ARS-UCD1.3”:

“The gene sets for the *Bos taurus* assemblies, UOA_Wagyu_1 (GCA_040286185.1), were generated using the Ensembl Gene Annotation system (PMID: 27337980). Annotations were primarily created by aligning short- and long-read transcriptomic data to the genome...”

2. Where is the CENP-A raw data from? There are no details in methods nor Supplementary information/methods

We thank the reviewer for highlighting this and have added details to the methods at line 709.

“We obtained CENP-A CUT&RUN data from GSE262830 and followed the CENP-A enrichment analysis previously detailed in the Wagyu Y chromosome work.”

3. Line 175 – What does CDR stand for?

We have added “centromeric dip region” to this sentence at line 280.

4. Figure 6 is really interesting, comparing the different algorithms to detect SVs, but it’s barely mentioned in the main text nor in the Supplementary information. Could the authors maybe expand this analysis? Do you see a pattern between those SVs called for all algorithms to those unique to one/two? And which ones are the ones then analysed in the rest of the panels of this figure?

We appreciate the reviewer’s interest in figure 6. The purpose of this analysis was to generate a robust list of candidate SVs rather than to evaluate any systematic biases present in each caller. We based this approach from Wu et al., (2021) who recommended using multiple callers to identify SVs and Dai et al., (2023) who followed this approach for cattle. Moreover, to ensure high quality SV, we added the extra criterion of the SV being present in at least 2 samples. Any SV identified by at least 2 callers in at least 2 samples was retained for further analysis and is presented in the remainder of figure 6.

References:

Wu, Z., Jiang, Z., Li, T. et al. Structural variants in the Chinese population and their impact on phenotypes, diseases and population adaptation. *Nat Commun* 12, 6501 (2021). <https://doi.org/10.1038/s41467-021-26856-x>

Dai, X., Bian, P., Hu, D., Luo, F. et al. A Chinese indicine pangenome reveals a wealth of novel structural variants introgressed from other *Bos* species. *Genome research*, 33(8), 1284–1298 (2023). <https://doi.org/10.1101/gr.277481.122>

5. For the BTAX evolution discussion, it might be worth looking into <https://www.mdpi.com/2073-4425/8/9/216>

We appreciate the reviewer's suggestion regarding the discussion on BTAX evolution. We have added the following statement at lines 483-485:

“Analysis of Cetartiodactyla species showed the general structure of X chromosomes and centromere positioning in pigs, alpacas, and whales are conserved whereas major centromere repositioning occurred in ruminants.”

6. There are track changes and comments still on the Supplementary Methods section, with some good points that should be addressed.

We thank the reviewer for noticing the comments. We have removed them, and the points have been addressed.

Reviewer #1 (Remarks on code availability):

Not all code is available, I could only review the github repository to quality control the assembly.

Our apology for the code unavailability. It was due to permission setting, and we have made the github repository publicly available now.

Some lines of code to run several tools are provided in the Supplementary Methods, and they work fine.

We thank the reviewer for checking this.

Reviewer #2 (Remarks to the Author):

The paper by Pineda et al. describes a haplotype-resolved assembly of a Tuli × Wagyu F1 hybrid cow, constructed using multiple sequencing technologies (PacBio HiFi, ONT—including ultra-long reads—Hi-C, and Illumina). Illumina reads were also generated for both parents to support the phasing process. The genome assembly comprises five chromosomes with telomere-to-telomere (T2T) status (i.e., chromosome-scale contigs with a resolved centromere and telomeric repeats at both ends), including the X chromosome. In addition, it includes 15 other chromosome-scale contigs that do not meet the T2T criteria, along with other chromosomal contigs and a total of 17 gaps. This assembly enables, for the first time, a detailed description of the structure of the X chromosome—specifically, the position and composition of its centromere. Based on orthologous positions of neighboring genes in related genomes and sequence composition analysis of the predicted centromere, the authors conclude that it represents a neocentromere that emerged in the ruminant lineage. The authors also demonstrate how this assembly makes it possible to describe the composition and structure of autosomal centromeres in cattle, to clarify the nature of the conservation of the pseudoautosomal region between cattle and primates, to expand the cattle gene catalog, and to characterize the structural variations in the Wagyu breed. Overall, the results presented here represent a significant amount of work using state-of-the-art sequencing technologies and methodologies. However, both the work itself and, perhaps more importantly, the way the results are presented raise some major concerns,

which we detail below.

Major concerns:

1/ X chromosome neocentromere: This work identifies, for the first time in cattle, the location of the centromere of the X chromosome. More strikingly, the authors indicate that it is a neocentromere, the centromere is not found at the expected location given the orthology map with related species (water buffalo, sheep, goat, human and chimpanzee). This observation enables the authors to further define the specific architectural features of the bovine X chromosome centromere which are summarized in the abstract (low CENP-signal, low methylation, among others) in contrast to the autosomal centromeres. This raises a number of questions.

The authors report a neocentromere on the cattle X chromosome, identified based on sequence features (cluster of inverted repeats) but lacking a CENP-A signal. To the best of our knowledge, there is no example in the literature of (neo)centromeres lacking an epigenetic signal such as CENP-A binding. This is supported by ref 10 (first key point), ref 18 in the manuscript, and ref 40 “From fission yeast to humans, the histone H3 variant CENP-A was demonstrated to be the epigenetic mark for centromere identity and function”. Since this result is central to the paper, the introduction should present in a more structured way current knowledge on neocentromere formation. Citing ref 20 (line 63) the authors mention “a different type of neocentromere (ENC)... displaying few of the centromere landmarks” suggesting that the cattle X-chromosome neocentromere described in the manuscript belongs to this class. But ref 20 precisely describes a neo-centromere that lacks specific sequence features (centromeric satellites) but is associated to a CENP-A signal.

We are not disputing the result reported by the authors, but the way it is presented is unsatisfactory, and as a result, the arguments put forward to support this thesis do not seem entirely convincing.

We thank the reviewer for the suggestion to improve the writing of the Introduction on centromere/neo-centromere and association with repeats, methylation and CENP-A. We have restructured the Introduction accordingly at lines 58-98.

“Centromeres are essential for the correct partitioning of chromosomes during mitosis and meiosis. They are typically characterised by tandem repeats organized into higher order repeats (HOR), enrichment of the centromere protein A (CENP-A) marking the site of kinetochore assembly, and dense methylation, except for regions of hypomethylation within CENP-A core domains. In rare cases, the centromere can reposition to an ectopic location, resulting in the formation of a neocentromere. Two types of neocentromere formation have been described: the first type has been observed in human clinical cases and is known as the human neocentromere (HN). HNs can lead to severe disease such as congenital abnormalities and cancer. Moreover, unlike canonical centromeres, HNs do not contain the alpha satellite DNA and are only found in single copy sequences. However, like canonical centromeres, they are still identified by their CENP-A signal; although low levels are found along the chromosome. Given their often deleterious nature, HNs are not fixed within populations. The second type has been observed in eight mammalian species (horse, donkey, plains and imperial zebra, Bornean and Sumatran orangutan, squirrel monkey, and macaque) and are believed to have arisen via a non-deleterious centromere relocation that has then become fixed within the population. This type of neocentromere is known as an

"evolutionarily new centromere" (ENC). These ENCs are generally characterised by either tandem repeats or non-repetitive sequences and are associated with CENP-A. The methylation status of these ENCs is not known."

We are glad the reviewer is not disputing the results as we have observed high CENP-A signal association with Bovine SatIII in autosomal centromeres. The generally low CENP-A signal in cattle sex chromosomes has been observed before in the Wagyu Y chromosome paper. We have discussed the features of the cattle X centromere at lines 509-517 as follows:

"In the literature for mammals, we could only find eight species with cases of ENC. These ENCs are generally associated with CENP-A and therefore, our finding of BTAX neocentromere low CENP-A is unprecedented. CENP-A in cattle is strongly associated with SatIII. The centromere of BTAX lacks any bovine satellites in it and as such CENP-A may only be loosely bound to the centromeric inverted repeats. Future experiment with anti-CENP-A antibody staining will confirm the active X neocentromere. We observed low methylation signal in the cattle X centromere, but other mammalian ENCs have no methylation data on their neocentromeres for a comparison. The large inverted repeat feature of cattle X neocentromere is not shared by other mammalian ENCs."

2/ T2T assembly: The quality of the assembly presented in the paper raises a number of issues.

The current assembly achieves a T2T status only for 5 out of 30 chromosomes, in contrast to the previously published T2T assemblies of two ruminant species (ref 31 and 32 for sheep and goat respectively), for which a T2T status was obtained for all chromosomes. The authors do not discuss the reason that might explain the differences. The reason is most probably related to the limited sequencing depth of ultralong reads (18.3X) in sharp contrast to the assembly protocol for the sheep and goat T2T assemblies (189X and 114X of ultralong ONT reads for sheep and goat respectively).

This is probably the motivation for testing various combinations of input data, parameters and software for the assembly process. But the comparison with other approaches is never discussed (mentioned only briefly, lines 78-79 main manuscript "to contrast strengths/weaknesses of previously described approaches") nor is the rationale for testing the combinations. A discussion about the failure to assemble all chromosomes at the T2T stage is, at the very least, necessary in our view.

We appreciate the reviewer's comment in the genome assembly of the Wagyu cattle, and we have added the following in Discussion at lines 541-548.

"The Wagyu genome presented here is not T2T for all chromosomes likely because of insufficient coverage of ultra-long reads (only 18.3x of >100 kb ONT), and the high sequence similarity of cattle centromeric repeats and rDNA has led to unresolved tangles in the genome graphs. These constraints make it harder to completely resolve the cattle genome despite numerous attempts in genome assembly including the use of HERRO corrected reads to boost sequence coverage of HiFi-like reads. Future efforts to complete cattle genome should be directed at increasing ultra-long reads and manually disentangling genome graph, assuming that few unresolved tangles remain post assembly."

The rationale for testing different combination of input data and assembly parameters was to obtain the most contiguous assemblies that have the most T2T chromosomes. The comparison of different assembly methods is given in the Supplementary Info section ‘Selecting the best assembly run’ and we have added the rationale.

Similar to efforts to assemble the human genome, the consortium decided to publish the T2T X chromosome first because this is of interest to the research community given that mammalian sex chromosomes have been the hardest to assemble and tend to be highly fragmented until recent innovation in sequencing technology and assembly algorithm.

In the Genome assembly and polishing section of the Methodology, it is explained that, when scaffolding with HiC data, no contigs were scaffolded correctly. This is difficult to understand without additional explanation, and this should be at least discussed.

Thank you for the reviewer's comment. We used Verkko to assemble our genomes, which incorporated ultralong reads to scaffold and bridge contigs into draft assemblies. Although, we also tried scaffolding with Hi-C data using YaHS, two chromosomes were erroneously joined together, which we removed. Given the high level of assembly completeness after Verkko, we have decided not to use the Hi-C/YaHS scaffolding result because it added no value. We have added as follows in lines 626-630:

“Verkko includes a scaffolding step that incorporates ONT reads to connect and bridge contigs.”

“Although, scaffolding with Hi-C data using YaHS software was attempted, two chromosomes (BTA2 and BTA11) were erroneously joined together, hence this was excluded due to misassembly.”

In the “Wagyu-specific genome improved mapping...” of the Results, it is explained that the mapping rate to the UOA_Wagyu_1_Y was lower than to the ARS-UCD2.0, but that this mapping rate becomes higher after removing centromeric sequences. It is not reasonable to think that, for a given assembly, some regions should be carefully masked before performing mapping (masking a genome is not recommended for mapping <https://github.com/lh3/minimap2/issues/654>). This low mapping rate of the T2T assembly contrasts with the increased mapping rate observed for the goat (ref 32 Supplementary Fig. 10) and a reduced error rate for sheep (ref 31 Fig 4a)

We thank the reviewer for their thoughtful response to the mapping rate. We have revised the methods and results for this section. We did not mask any region of the genome prior to mapping and agree this would be problematic and onerous to require this. Re-calculation of mapping rates to placed chromosomes using Qualimap bamqc produced results that were in line with expectation of improved mapping rates to the Wagyu reference. We have amended the results at lines 396-399 as:

“Using long reads from the 20 Wagyu samples in this study, we observed an average mapping rate of 86.84% when mapped to ARS-UCD2.0 (Supplementary Table 21). In contrast, we observed an average mapping rate of 94.80% when aligning to UOA_Wagyu_1_Y.”

We have also amended the methods at lines 812-814 as:

“We used a bed file with the coordinates of each placed chromosome for the respective genome with Qualimap bamqc to determine mapping rate for a given sample and reference genome.”

Overall, whether regarding the assembly process or the elements supporting the neocentromere hypothesis, the relevant information is very difficult to locate in the paper. We have reorganised the structure of the paper to make it easier to locate the information. Please refer to lines 110-155, 186-190, 210-212, 222-231, 254-255, 281-283, 374-377, 394-400 and 406-413.

This brings us to the main criticism of the paper.

3/ Manuscript organization: The way the manuscript is structured poses a major problem for the reader. Information related to the results and methods appears in different sections with no apparent logic. We give an example to illustrate this point. For the assembly process, the “various combinations of input sequence data...” are first mentioned in the Results section (line 78), but those combinations are only understandable by gathering information from the Methodology section, the Supplementary notes, Supplementary Methods and the Supplementary Table 1. The reader has to make his own calculation to understand how the 228.8X of ONT data are used (72.8X of R9.4.1 (line 426, Main Manuscript) 57X of R10.4.1 >=10kb (line 429 MM and line 401 Supplementary Methods), 99X of R10.4.1 >= 30kb (line 404, SM)).

We manuscript organization has now been restructured, see lines 110-155. The reader no longer has to make his/her own calculation as this is now provided. As this is a Nat Comms paper, it is typical more detail is included in Supplementary.

We note that sequencing depth information for the sheep and goat T2T assemblies (ref 31 and 32), for all the technologies, are available in the very first sentence of the results section, which immediately makes it possible to know them.

We appreciate your note and have added the sequencing depth in the result section of the genome assembly at lines 110-155.

“A male calf from a Wagyu dam and Tuli sire cross was selected for haplotype-resolved genome assembly. We generated 228.8x ONT long-read coverage including 18.3x ONT ultra-long (>100 kb), 58.1x PacBio HiFi, ~421 million Proximo Hi-C reads, 81.8x F1 Illumina short-read, and 78x parental Illumina short-read coverage (Supplementary Table 27).”

“The sequencing coverage of the assembly selected was 58.1x of PacBio HiFi reads, 57x of HERRO corrected ONT reads as input along HiFi, 121x coverage of ONT reads and 18.3 x ONT ultra-long reads (>100kb).”

For all the above reasons, we believe that this manuscript is not yet suitable for publication and requires an extensive rewrite.

We have reorganised the structure of the paper as detailed above.

Below, we provide some examples of problematic points, following the order in which they appear in the manuscript.

Minor concerns:

Introduction

Line 42: ref 8 is not appropriate. The paper deals with telomeres, but nothing is said about the challenge they pose in assembly

We apologise that ref 8 was a mistake. We have updated the references with Pineda et al., (2024), which showed incomplete telomeres and centromeres of different assemblies. For example, telomeres can be misplaced in the assembly, such as the case of chromosome 1 of water buffalo UOA_WB_1.

Line 45: ref 10 is not appropriate for the specific problems caused by repetitive sequences for the assembly of allosomes

Our apology, ref 10 has now been removed.

Line 46 “X chromosome is of particular importance as it is found in all mammalian cells”. Are there examples of chromosomes not appearing in all mammalian cells?

We have rewritten the sentence at lines 47-49 as follows:

"The X chromosome is important as it contains several genes with important roles in a multitude of traits, such as intelligence and mental function, intellectual disability, reproduction, and milk production."

Line 47 Emphasis is made on intelligence, intellectual disability, mental function (with reference 12 appearing twice). For the reader, such emphasis is difficult to understand since this aspect is not mentioned later on in the paper

We appreciate the reviewer’s comments and have rephrased this sentence to try and reduce the emphasis on these traits and make it clearer that these were intended as examples of traits that may be of interest. We have revised it in lines 47-49, which is same statement as response to your previous comment.

Lines 53-64 The description is not appropriate. It is mentioned that ENC for “evolutionary centromere” is a different type of neocentromere that displays few of the centromere landmarks, without providing a description of the different types (or the other type) of neocentromeres.

We acknowledge the reviewer’s note and have revised our statement as follows:

There are two types, human neocentromeres (HNs) and evolutionary new centromeres (ENCs). This section has been extensively rewritten in lines 63-98, which is also given above in response to one of your comments.

Lines 72-73 The T2T assemblies sheep and goat should be mentioned here (ref 31 and 32 respectively)

We have added the sheep and the goats T2T assemblies.

Results

Line 84 and line 97. The unplaced scaffolds are mentioned without any reference to the section of the Methods explaining the scaffolding process.

We have revised the Methods section at line 626-627 and explained that Verkko has an integrated scaffolding process.

Line 114 “rDNA arrays prevented the complete resolution of BTA4”, How this array was detected on BTA4 is not explained. A reference to the corresponding section in the Supplementary Methods should be added, and this section should mention BTA4.

We value the reviewer’s comment and have revised the sentence in the Results at line 203-204 as follows:

“Notably, rDNA arrays prevented the complete resolution of BTA4_(Supplementary Information).”

In the Supplementary Information under section ‘rDNA copies’, we have revised it as follows:

“The rDNA units found in unplaced scaffolds were also found in the string graphs connecting to BTA3, BTA4, BTA11 and BTA25 (Figure 1f). This was done by identifying the nodes corresponding to the scaffolds. Although BTA4 has a telomere on the p-arm and without gaps, the presence of an rDNA at the q-arm and unplaced rDNA-containing nodes in the string graph suggest that the rDNA prevented the assembly of a T2T chromosome.”

Line 168 A reference to CARP (ref 70) should appear here the first time it is mentioned.

We thank the reviewer for noticing. We have added the reference for CARP.

Line 171 The TE described (Bisbis, LTR11B_BT) are most certainly not 849kb and 477kb long respectively, the sentence should probably be rephrased

We have rephrased it in line 275 as follows:

“The largest BTAX centromere TE is Bisbis-1.8, covering a total of 849 kb of the BTAX centromere, followed by LTR11B_BT (477 kb) and BosInd-1.103 (300 kb).”

Line 187 A reference to the Supplementary Methods section where expected CpG count is defined should be provided

We appreciate the reviewer highlighting this absence and have added a reference at line 293-294.

“The relationship between the observed CpG count and the expected CpG count was linear (Pearson’s $R = 0.95$, $P\text{-value} = 1.01e-18$) (Fig. 4b) (Supplementary Methods).”

Line 264 The gene annotation process is never described, but the reader understands that it was made using the Ensembl pipeline

We have made reference to this annotation pipeline clearer in the main text and supplementary information.

In the main text at line 186-190:

“The genome annotation was performed by EBI using Ensembl Gene Annotation system, which integrated both short- and long-read transcriptomic data with protein homology

model, with emphasis for gene models with transcriptomic data support (Supplementary Information).”

In the Supplementary under section “Gene annotations of UOA_Wagyu_1 and its comparison with ARS-UCD2.0 and ARS-UCD1.3”:

“The gene sets for the Bos taurus assemblies, UOA_Wagyu_1 (GCA_040286185.1), were generated using the Ensembl Gene Annotation system (PMID: 27337980). Annotations were primarily created by aligning short- and long-read transcriptomic data to the genome...”

Line 280 The reduced mapping rate issue has already been mentioned above
We have addressed the comment above.

Line 310 ref 30 is not related to BOVA2 SINE.
We thank the reviewer. We have removed reference 30.

Discussion

Line 389 The proposed relationship between Wagyu SV hotspots targeting olfactory transduction associated genes and breed’s superior feed efficiency and marbling is an overstatement that is not supported by the associated references (ref 53 and 54)
We have removed the sentence.

Methodology

Line 429 The correction reads using HERRO of a first batch of ONT reads (57 x R10.4.1 with minimum read length of 10kb), appears here for the first time, with no apparent reason
We have revised the sentence in line 611-616 to the following:

“A subset of ONT long reads (~57x coverage, R10.4.1) was corrected using HERRO (Haplotype-aware ERRor cORrection). Prior to correction, the reads were filtered to retain only those with a minimum length of 10 kb and a minimum average read quality of QV10. The HERRO-corrected ONT reads were used as additional high-accuracy “HiFi” input because of its improved base-level accuracy.”

Line 446 ref 24 is inaccurate here but correctly placed line 448
We thank the reviewer for noticing, we have removed the incorrect reference placement.

Reviewer #2 (Remarks on code availability):

Git repository with code associated to the manuscript

Assembly initial QC: https://github.com/plnspineda/assembly_initialqc

The code is not usable for the submitted assemblies without substantial code modification (either X or Y is absent from the submitted haplotypes)

LR-variantcaller pipeline <https://github.com/DaviesCentreInformatics/LR-variantCaller>

“is designed to be run on the University of Adelaide's HPC, Phoenix” as explained in the README.md file. This code is intended for internal use.

Cattle T2T code repository https://github.com/DaviesCentreInformatics/Cattle_T2TX

Contains essentially the R code to produce the figures. While this is very useful, it does not make it possible to understand and replicate the various steps of the analysis. When the

code is provided (Testis Gene expression), the very large number of absolute paths highlights that the purpose of sharing it is not to enable readers to reproduce the analyses. And in some cases, it simply refers back to the paper https://github.com/DaviesCentreInformatics/Cattle_T2TX/tree/main/assembly_evaluation

We have revised the description on how to use the analysis code. How the users may set up equivalent input to replace our paths is given. This is not a paper on software code and hence, we could only try to make the description of our analysis code as generic as possible. Some efforts by the users are needed in order to replicate similar analyses on their high performance computers.

Reviewer #3 (Remarks to the Author):

This manuscript reports on cattle T2T chromosomes including a T2T X chromosome. Authors use this resource to look into the evolution of centromeres. Methods and approaches appear sound. Overall this appears as a useful manuscript for the bovine genomics community but it is very descriptive and contains a substantial portion of text and analyses that appear off topic.

Authors compare genes annotated on the Wagyu X, Y, and PAR with those annotated in other species. Would be good to clarify early on how the new Wagyu genome was annotated (e.g., in the first Results paragraph), and if the annotation is public. I was unable to find Ensembl entries for the Ensembl IDs mentioned in e.g., Table S15
We have made reference to the wagyu annotation clearer in the main text and supplementary information.

In the main text at line 186-190:

“The genome annotation was performed by EBI using Ensembl Gene Annotation system, which integrated both short- and long-read transcriptomic data with protein homology model, with emphasis for gene models with transcriptomic data support (Supplementary Information).”

In the supplementary under section “Gene annotations of UOA_Wagyu_1 and its comparison with ARS-UCD2.0 and ARS-UCD1.3”:

“The gene sets for the Bos taurus assemblies, UOA_Wagyu_1 (GCA_040286185.1), were generated using the Ensembl Gene Annotation system (PMID: 27337980). Annotations were primarily created by aligning short- and long-read transcriptomic data to the genome...”

We have included links to the Ensembl annotation in the methods at line 676-679.

“Annotation files of the UOA_Wagyu_1 are available through Ensembl, accessible at https://ftp.ebi.ac.uk/pub/ensemblorganisms/Bos_taurus/GCA_040286185.1/ensembl/genes_et/2024_11/. Alternatively, the annotation can be viewed at Ensembl Genome Browser <https://beta.ensembl.org/species/6d661feb-0bc2-4b4f-baa4-9e1e7a0c764f>.”

population genetic analyses presented in lines 278 - 326, and lines 380-391 are off-topic

given the title of the manuscript. Moreover, I may have missed it, but couldn't find anything about the short read sequencing and variant calling in methods.

We appreciate the reviewer's comments. The analyses presented are quite common for newly assembled genomes. For example, Dai et al., (2023) assembled a Chinese indicine genome, investigated SVs and identified genes likely impacted by these SVs and how they may influence phenotypic traits of the breed. Yang et al., (2023) generated a T2T assembly of a Han Chinese male and investigated what impact this new population-specific genome has on the ability to detect rare and population-specific variants.

Our apology for missing the short read variant calling methods. We have included it at line 796-799.

"Publicly available Wagyu short reads were downloaded from NCBI and mapped to UOA_Wagyu_1 genome for variant calling using the method described in Angus and Brahman assembly work (Supplementary Table 22)."

References:

Dai, X., Bian, P., Hu, D., Luo, F. et al. A Chinese indicine pangenome reveals a wealth of novel structural variants introgressed from other Bos species. *Genome research*, 33(8), 1284–1298 (2023). <https://doi.org/10.1101/gr.277481.122>

Yang, C., Zhou, Y., Song, Y. et al. The complete and fully-phased diploid genome of a male Han Chinese. *Cell Res* 33, 745–761 (2023). <https://doi.org/10.1038/s41422-023-00849-5>

Low, W.Y. *et al.* Haplotype-resolved genomes provide insights into structural variation and gene content in Angus and Brahman cattle. *Nature Communications* **11**, 1-14 (2020)

Given the very high transcriptional complexity of testes, I wonder if the fact that genes are expressed in testis indicates indeed that they play a role in fertility (line 34, lines 393ff)
The mammalian X chromosome is enriched in genes expressed in spermatogenesis (reviewed in Vockel et al 2019, Human Genetics). In human, male infertility is linked to the X chromosome. Human and cattle are diverged ~94 million years ago. Our findings that the 18 X-PAR have conserved testes expression between cattle, human and primates and all X centromeric cattle genes are expressed in testes suggest these genes as worthwhile candidates genes for further investigation into their potential roles in male fertility. As our genome is newly annotated, it will require a much larger efforts to map and compare expression of other tissues with testes to discover potential testes specific or enriched testes genes on the X chromosome.

Reference:

Vockel, M., Riera-Escamilla, A., Tüttelmann, F. et al. The X chromosome and male infertility. *Hum Genet* 140, 203–215 (2021). <https://doi.org/10.1007/s00439-019-02101-w>

please use consistent terms; e.g., "p arm" vs. "p-arm"

We appreciate the reviewer's notes. We have revised the terms for consistency. For example, we have used p-arm and q-arm throughout.

Here are a couple of line-specific comments:

line 23: "The cattle genome is crucial for understanding ruminant biology" - this is a very unspecific sentence. Are you referring to a completely annotated reference sequence? This is a general statement on how the biology of a species or group of species can be better understood using their genetic makeup. The words "ruminant biology" were used in the original cattle genome paper in Science in 2009.

Reference:

The Bovine Genome Sequencing and Analysis Consortium et al. The Genome Sequence of Taurine Cattle: A Window to Ruminant Biology and Evolution. Science 324, 522-528 (2009). <https://doi.org/10.1126/science.1169588>

lines 39-41: "bovine" & "bovids" - the cited references refer to cattle, not bovids. We have edited the term and used cattle instead.

line 66: "complete gapless" - complete can be deleted
We have removed "complete".

line 77: would be good to present at least some information about the data used for the assembly process, as well as the DNA source (sex and tissue). Otherwise readers need to look that up in the methods.

We thank the reviewer for the suggestion. We have added additional information in the Results sections of the genome assembly at line 110-155.

"A male calf from a Wagyu dam and Tuli sire cross was selected for haplotype-resolved genome assembly. We generated 228.8x ONT long-read coverage including 18.3x ONT ultra-long (>100 kb), 58.1x PacBio HiFi, ~421 million Proximo Hi-C reads, 81.8x F1 Illumina short-read, and 78x parental Illumina short-read coverage (Supplementary Table 27)."

l. 126: not clear if this was estimated from the HiFi reads, or from an assembly built from HiFi reads

The statement in line 218 "average telomere length was 6.8 kb" were from the assembled genome while the next sentence was estimated from PacBio HiFi reads. We have added "For assembled chromosomes in the genome, the average telomere length was 6.8 kb." to avoid confusion.

line 175: please define CDR

We have added the definition of CDR in line 280,

"Centromeres are known to be epigenetically defined, with recent work revealing a centromeric dip region (CDR) where CpG methylation decreases and coincides with the presence of CENP-A."

line 177: is there an "in" missing?

We have edited to include "in" at line 283 as follows:

"While high CENP-A and CDRs were not observed in BTAX centromere..."

line 227: similar to what? please clarify

We have revised the statement at line 336 for clarification to:

“The X-Y PAR region contains 31 similar genes to each other aside from the PRP gene (Supplementary Table 11-12)”

line 248: the number of 17k testis-expressed genes is much lower than the 20k reported in <https://doi.org/10.1038/s41467-024-44935-7>

Thanks for the comments. The difference is due to our method for RNA-seq analysis is different, only five samples were common between ours and Mapel et al (2024), and the Wagyu annotation is also different. Since the other testis samples included in our work has lower sequencing depth, this could be the reason for lower number of expressed genes found.

Reference:

Mapel, X. M., Kadri, N. K., Leonard, A. S. et al. Molecular quantitative trait loci in reproductive tissues impact male fertility in cattle. *Nature communications*, 15(1), 674 (2024). <https://doi.org/10.1038/s41467-024-44935-7>

line 267: are the 738 newly identified protein-coding genes functional, i.e., can they be confirmed with e.g., RNAseq?

We appreciate the reviewer’s comments regarding the newly identified genes. We agree this would be interesting to investigate in a future study, but we chose to focus the work on the complete X chromosome and have limited the gene expression analysis to the testes. Additionally, we have pointed out that the gene annotation pipeline used by EBI emphasized the use of transcriptomic data to support the gene models.

line 279ff: authors need to provide more details at the beginning of the paragraph what they did. How many samples were considered? average coverage of what?

We have revised the text in this paragraph at line 394-396 to improve the clarity of what was done.

“As genomes become more complete, the mapping rate of sequencing reads tends to improve. To assess this, we investigated the mapping rates of ~10x long-reads and ~11-20x short-reads generated from Wagyu animals. Using long reads from the 20 samples in this study, we observed...”

line 292: same as above. More details needed. What is the "MGC method"? Are these 20 samples the same as used above for the alignment and variant calling? Did you consider only SVs identify by all 6 callers? Kind of odd that most SVs appear to be present in at least 3 samples, wouldn't a large fraction of singletons be expected? did you use any functional data to verify TSS?

We appreciate the reviewer’s comments and have sought to address them as follows:

We have revised the text at the start of this section to provide an overview of the MCG method at line 406-411.

“The MCG method works by first selecting the individual that is the most genetically representative of the population e.g. a commonly used sire. It then selects the next sample that captures the most genetic variation not observed in the first sample, with this process repeated to capture more genetic diversity with less samples. We employed the MCG method to reduce the number of samples needed to capture the most genetic variation within our research budget.”

The MCG method was how we chose which 20 animals to use for long-read sequencing and therefore are also the samples that we used for long read variant-calling mainly for SV discovery.

No, we did not use only SVs identified by all callers. We have outlined in the Methods how we selected SVs for further analysis; in short, we considered any SVs that were found in at least 2 samples by at least 2 callers as valid SVs.

“SVs from these four tools were merged with SURVIVOR (v1.0.7), with only SVs identified in at least two samples by at least two callers being retained for further analysis...”

We agree with the reviewer’s assessment that we would expect most SVs to be singletons, and this is why we chose to only consider SVs present in at least 2 samples and identified by at least 2 callers. This strict criteria for inclusion were chosen to try and limit the false positive rate.

The TSSs are annotated by Annovar and the methodology we used for this analysis is similar to previous published e.g. Dai et al., 2023.

Reference:

Dai, X., Bian, P., Hu, D., Luo, F. et al. A Chinese indicine pangenome reveals a wealth of novel structural variants introgressed from other Bos species. *Genome research*, 33(8), 1284–1298 (2023). <https://doi.org/10.1101/gr.277481.122>

line 413: this appears as an overly bold claim which is not substantiated by any results. How will the catalogue of SVs inform future GWAS and genomic prediction?

We have edited this sentence at line 591-597 to more accurately reflect the current situation with cattle SVs.

“We produced a high-confidence catalogue of cattle SVs that we anticipate will be useful for global efforts in investigation of the incorporation of this important class of variant in future genome wide association studies and genomic prediction work to select for agriculturally important traits.”

The SVs produced in this work are part of the Bovine Long Read Consortium (BLRC) (Nguyen et al 2023, GSE). It is common knowledge for members of this consortium that SVs are likely important for GWAS and genomic prediction, but the reviewer is right that this has not been demonstrated in this work. We note the following sentences from the BLRC for your reference.

‘Either alone, or combined with imputed SNPs and INDEL, this would enable population-scale and GWAS with SV to determine the impact of SV on quantitative trait phenotypes as well as Mendelian traits. Furthermore, we can anticipate that the increasing availability of these resources in genomic prediction settings for a range of traits will deliver positive impacts for livestock breeding.’

Reference:

Nguyen, T.V., Vander Jagt, C.J., Wang, J. et al. In it for the long run: perspectives on exploiting long-read sequencing in livestock for population scale studies of structural variants. *Genet Sel Evol* 55, 9 (2023). <https://doi.org/10.1186/s12711-023-00783-5>

line 419. Clarify why the F1 is a hybrid. Are the parents from different species?

The F1 refers to African Tuli crossed with Wagyu. Similar to our previous work in genome assemblies that used Trio binning method e.g. Angus x Brahman in Low et al. 2020 in *Nat Comms*, we considered the offspring as F1 hybrid. More details in Supplementary Info under section ‘Sequencing of the trio (Tuli x Wagyu)’.

Reference:

Low, W.Y., Tearle, R., Liu, R. et al. Haplotype-resolved genomes provide insights into structural variation and gene content in Angus and Brahman cattle. *Nat Commun* 11, 2071 (2020). <https://doi.org/10.1038/s41467-020-15848-y>

line 430: ref for Herro needed?

Thank you. We have added the reference accordingly.

line 442: ref for minimap needed?

We have added the reference accordingly.

there are a couple of author comments left in the Supplementary Notes document (e.g., at page 28, or for Supplementary Figure 4) which the authors should remove
We thank the reviewer for noticing. We have removed the comments.

RESPONSE TO REVIEWERS' COMMENTS

Reviewer #1 (Remarks to the Author):

The authors have addressed all my comments. They have expanded the main manuscript giving enough context in each section and included more details on BTAX and CENP-A analyses as I requested.

Reviewer #1 (Remarks on code availability):

Code now is fully available, but I didn't test it. However, the data is not in the github repo and I'm not sure how reproducible it would be.

The data is available as Source Data and also described in the Data Availability section.

Reviewer #2 (Remarks to the Author):

The authors have thoroughly addressed all the comments. I have only a few minor additional comments.

1. Unless I have overlooked it, the way telomeres are detected in the assembly is not described (line 219 refers to the supplementary information, most certainly to the paragraph Telomere length variations, but how the units are detected is not explained). A description or at least a reference (Pineda et al. 2024 ?) should be provided.

The telomere units were counted using Vertebrates Genomes Project's (VGP) script 'telomere_analysis.sh' which is described in the Supplementary Methods under the "Assembly evaluation" subheadings with referencing to Rhie et al., 2021.

2. Lines 272-276. A reference for BisBis-1.8 (3657bp, Dfam or RepBase ?), LTR11B_BT (1053bp, RepBase ?) and BosInd-1.103 (8366bp, Dfam or RepBase ?) would help clarify the manner in which these TEs span the BTAX centromere.

We have added the library used for TE analysis at Line 244 "The largest BTAX centromere TE from the RepBase (v29.04) library is Bisbis-1.8".

Reviewer #2 (Remarks on code availability):

The repository for the code that enables reproduction of all the analyses has been considerably and comprehensively improved. I believe this will be of great value for the

readers.

Reviewer #3 (Remarks to the Author):

Authors have addressed most of my comments and made appropriate changes in the manuscript.

However, the population-genetic analyses still appear off-topic and don't contribute much. The title of the manuscript and the abstract (and to a large extent also the introduction) indicate that the manuscript addresses the cattle sex chromosomes and centromere evolution. The SV analyses from 20 long-read sequenced samples aren't mentioned at all. Moreover, these SV analyses don't contribute to the characterisation of centromeres or cattle sex chromosomes. The statement that such analyses are quite common is odd as the references cited (Dai et al., Yang et al.) focus on improved assemblies whereas the current manuscript deals with centromere evolution. I stand by my initial assessment that analyses presented in lines 405-462 are off topic.

We have added text to the abstract (Line 28-29), introduction (Line 89-98) and discussion (Line 525-529) to make the connection between the population genetic analysis and the rest of the paper clearer.